# Emotions Where Art Thou: Understanding and Characterizing the Emotional Latent Space of Large Language Models

**Benjamin Reichman**
Georgia Institute of Technology
`bzr@gatech.edu`

**Adar Avsian**
Georgia Institute of Technology
`aavsian3@gatech.edu`

**Larry Heck**
Georgia Institute of Technology
`larryheck@gatech.edu`

## Abstract

This work investigates how large language models (LLMs) internally represent emotion by analyzing the geometry of their hidden-state space. The paper identifies a low-dimensional emotional manifold and shows that emotional representations are directionally encoded, distributed across layers, and aligned with interpretable dimensions. These structures are stable across depth and generalize to eight real-world emotion datasets spanning five languages. Cross-domain alignment yields low error and strong linear probe performance, indicating a universal emotional subspace. Within this space, internal emotion perception can be steered while preserving semantics using a learned intervention module, with especially strong control for basic emotions across languages. These findings reveal a consistent and manipulable affective geometry in LLMs and offer insight into how they internalize and process emotion.

## 1 Introduction

Large Language Models (LLMs) have become central tools for interacting with, analyzing, and generating human language. Their widespread deployment across domains has led to increasing interest in how they handle not just syntactic or semantic meaning, but also affective tone. Emotion is a fundamental part of language, shaping persuasion, social signaling, and narrative context. As such, understanding how LLMs process emotional content is essential for both interpretability and safe deployment.

The literature on affect in NLP has focused on sentiment analysis, a task where models classify inputs into discrete emotional or affective categories (Prabowo & Thelwall, 2009; Medhat et al., 2014; Wadawadagi & Pagi, 2020; Wankhade et al., 2022). While this demonstrates that LLMs can identify emotions, it offers little insight into how emotional meaning is represented internally. Classification accuracy is not equivalent to interpretability.

Other works have taken a behavioral view, exploring the emotional "intelligence" of LLMs. These include prompting models with hypothetical emotional scenarios and evaluating their responses (Wang et al., 2023), or probing how well they align with human judgments in affective tone (Huang et al., 2024; Zhao et al., 2023). Though these studies suggest some degree of affective sensitivity, they focus on testing outputs rather than investigating internal mechanisms of the LLM.

Recent work has also examined emotion manipulation and decoding. For instance, models have been used to map text to dimensional emotion ratings like valence-arousal-dominance (VAD) (Shah et al., 2023; Broekens et al., 2023), or to generate emotionally inflected language on demand (Reichman et al., 2025). LLMs have also been shown to be more likely to comply with emotionally framed requests (Vinay et al., 2024). These studies also treat emotion primarily as a label or generation condition—not a latent internal representation.

While there has been some work examining how LLMs respond to or generate emotional language, the structure of emotional representations within their hidden states remains relatively underexplored. Most prior approaches focus on output behavior or classification accuracy, with comparatively few efforts aimed at interpreting the internal geometry of emotion encoding. To advance the understanding of how emotions are represented in LLMs and how they influence LLM responses, this paper investigates the internal mechanisms of the LLM. Emotions are analyzed within LLM hidden states across layers, datasets, and languages.

Our contributions are as follows: (1) We extract a low-dimensional emotional subspace of the LLM and show that it captures interpretable, directionally encoded affective structure across LLM layers. (2) We demonstrate that this space generalizes across eight emotion datasets spanning five languages, with low alignment distortion and high cross-domain probe accuracy. (3) We introduce a learned steering module that manipulates internal emotional perception while preserving semantic content, with especially strong control over basic emotions. We find that emotional encoding is directional, distributed, and remarkably consistent across varied textual modalities. We also investigate the model's internal "psychology": how emotions are separated, aligned, and—critically—how they can be steered via targeted interventions.

## 2 RELATED WORKS

**Models of Emotions.** Psychological models of emotion are commonly categorized as either discrete or continuous. Discrete theories posit that emotions are fundamentally distinct categories—such as the six "basic" emotions proposed by (Ekman et al., 1999): anger, surprise, disgust, enjoyment, fear, and sadness. Other taxonomies expand this set, including more nuanced affective states (Plutchik, 1991).

In contrast, continuous models view emotions as points in a low-dimensional latent space. A widely used formulation is the valence-arousal-dominance (VAD) model (Mehrabian, 1996), where valence encodes hedonic tone, arousal measures intensity, and dominance reflects control or agency. Variants of this framework reduce or alter the axes (e.g., Russell's 2D circumplex (Russell, 1980)).

These representations offer an interpretive lens for analyzing learned emotion structure in LLMs: If models implicitly encode emotions in a geometric space, we may expect that certain latent directions align with these classic dimensions. Our work explores whether such a structure emerges naturally in the hidden-state geometry of LLMs trained without explicit emotional supervision.

Neuroscientific models of emotion offer a parallel debate. Localist theories posit that discrete emotions correspond to specific, anatomically distinct brain regions, while constructionist theories argue that emotions emerge from distributed, domain-general processes (Lindquist et al., 2012; Vytal & Hamann, 2010; Celeghin et al., 2017). Our results, particularly from ML-AURA (Section 4.3), support a constructionist-style interpretation in LLMs: emotional content is not localized to a small subset of units but is instead widely distributed across neurons and layers, with high separability emerging from overlapping, multi-purpose components.

**Emotions in Latent Space.** Recent work has investigated how LLMs interact with emotional text, often focusing on behavior or output-level mappings. For example, ChatGPT has shown the ability to map emotions to Valence-Arousal-Dominance (VAD) values (Broekens et al., 2023; Yongsatianchot et al., 2023), suggesting that emotion-relevant dimensions are accessible to the model. However, such studies do not analyze the internal structure or geometry of these latent representations.

Some prior work explicitly trains models to embed emotions into structured spaces, using classification objectives or external supervision. For instance, (Dathathri et al., 2019) and (Buechel et al., 2020) train models to map between emotion spaces. Similarly, (Wang & Zong, 2021) learns an emotion space from labeled data, shows clustering by valence, and demonstrates transferability across datasets. However, in all of these works, the emotion space is imposed or supervised, not emergent.

A growing line of work probes how pretrained models encode emotion. (Hollinsworth et al., 2024) show that valence is linearly embedded in contextual states, while (Zhang et al., 2023) find arousal and dominance less separable, though their analysis depends on encoder-only models and fixed affective lexicons. In contrast, we study decoder-only LLMs, seeking to recover emergent emotional structure directly from hidden-state geometry rather than imposing a psychological model.

Other studies have shown that LLMs exhibit strong zero-shot emotion classification performance across languages (Bianchi et al., 2022), though subsequent work notes that language-specific tuning is sometimes necessary for culturally grounded affect (De Bruyne et al., 2022). These findings suggest that emotion representations are at least partially transferable across linguistic domains—a hypothesis we test more directly through geometric alignment and projection-based analysis in Section 4.

## 3 METHODS

To understand how emotions are represented in LLMs, a variety of tools were used. This section outlines those methods and their theoretical grounding. Empirical findings from these analyses are presented in Sections 4 and 4.3.

**Centered-SVD.** We build on prior work showing that LLM hidden states lie on low-dimensional manifolds where semantic and syntactic properties are linearly recoverable (Aghajanyan et al., 2020; Hu et al., 2022; Lizzo & Heck, 2025). To isolate emotion-relevant subspaces, we apply singular value decomposition (SVD) to hidden state activations.

Emotion-relevant subspaces are extracted using centered singular value decomposition (SVD) on sentence-level hidden states. For each input, token activations are mean-pooled to obtain a single sentence representation; these vectors are stacked, centered, and decomposed via SVD to obtain orthogonal directions of variation. When emotional content is the dominant structured difference across the inputs, the leading components should capture the primary affective axes in the model's representation space and form the basis for the alignment, probing, and causal manipulation analyses described in later sections.

To interpret the semantic content of the principal components derived from the centered SVD, the relative ordering of emotion centroids are examined along each component. Component polarity is standardized by flipping signs when necessary to ensure consistent orientation across layers. This procedure yields a stable ranking of emotions along each axis, enabling cross-layer comparison and semantic interpretation of the leading components.

For qualitative visualization, we embed the mean-pooled hidden-state centroids into two dimensions using t-SNE, applied to the same activation representations used for SVD. This embedding provides a coarse, nonlinear view of cluster structure and complements the linear analyses conducted in the emotional subspace.

**Space Alignment.** We first assess cross-dataset similarity through centroid cosine similarity, measuring how consistently each emotion's direction aligns with its counterpart in the synthetic manifold; for each emotion, we compute the cosine between its dataset-specific centroid vector and the corresponding centroid in the synthetic space. Prior work has shown that latent spaces arising from related tasks often exhibit similar internal geometry, with relationships between them approximately rigid or linear up to rescaling and rotation (Moschella et al., 2023). While some approaches lift these spaces into anchor-relative representations to handle isometric variance, recent work demonstrates that direct alignment via linear or rigid transformations is often sufficient and easier to apply in practice (Lähner & Moeller, 2024). Following this approach, we use linear regression to align the emotional subspace derived from synthetic data with that derived from human-authored emotion datasets, allowing us to test whether the synthetic manifold reflects transferable emotional encodings or merely generation artifacts. $W^*$ is obtained by a least-squares fit over paired real–synthetic sentence-level activations.

$$W^* = \arg\min_W \| YW - X \|_F^2,$$

The resulting MSE between $X$ and $YW^*$ summarizes how well the synthetic manifold can be recovered from the real space. To characterize the learned transformation, we report its Frobenius norm (overall magnitude) and spectral flatness (isotropic versus anisotropic scaling), which help distinguish genuine geometric compatibility from alignment achieved through highly directional distortion.

To evaluate how consistently emotional structure is preserved across datasets, we complement the cosine-similarity and regression-based alignment metrics with a set of high-dimensional geometry measures that quantify relational distortion between emotional spaces. Whereas cosine similarity

evaluates directional agreement and regression error captures how well one space can be linearly mapped onto another, these distortion and stress metrics assess the internal geometry itself, that is, how faithfully pairwise relationships among emotions are preserved after alignment. They reveal whether two spaces share not only similar directions but also comparable distance structure, providing a stricter and more global test of representational equivalence.

**Geometry Preservation Metrics.** To evaluate how emotional structure is preserved across datasets, geometric deformation is quantified between emotional spaces using both classical stress measures and distortion metrics. All geometry-preservation metrics are computed over pairs of activation vectors from different datasets corresponding to the same emotion, comparing their pairwise distances in each dataset's latent space to the distances in the synthetic latent space. Stress scores quantify aggregate embedding discrepancy, while distortion metrics assess whether those discrepancies arise from uniform rescaling or more uneven, directionally biased deformation.

Stress-1, Stress-2, and Sammon stress measure how well one distance matrix can be embedded into another. Stress-1 computes the mean-squared discrepancy between distances, Stress-2 captures the same discrepancy without square-root normalization, and Sammon stress reweights errors by the inverse of the original distance, emphasizing preservation of local structure. Pairwise distances are defined by $D_{ij}^{(X)} = \|x_i - x_j\|_2$ and $D_{ij}^{(Y)} = \|y_i - y_j\|_2$. Stress-2 is then computed as

$$\frac{\sum_{i<j}(D_{ij}^{(H)} - D_{ij}^{(L)})^2}{\sum_{i<j}(D_{ij}^{(H)})^2} \tag{1}$$

Classical thresholds for low stress (e.g., Stress-1 < 0.1) were developed for low-dimensional maps (Kruskal, 1964) and do not directly apply to high-dimensional hidden-state spaces, so we interpret these values across models. Stress-1 and Sammon stress appear in the appendix.

Complementing these metrics, average distortion, $\ell_2$-distortion, and $\sigma$-distortion Chennuru Vankadara & von Luxburg (2018) quantify how distances change under the mapping itself. Average distortion measures the mean stretch factor across all cross-dataset emotion pairs (ideal≈1). Using the same pairwise distances as computed for stress, the stretch ratio is defined as

$$\rho_{ij} = \frac{D_{ij}^{(Y)}}{D_{ij}^{(X)} + \varepsilon} \tag{2}$$

$\ell_2$-distortion captures the squared deviation of stretch ratios from a single global scale, reflecting uneven expansion or contraction (ideal=0). $\sigma$-distortion measures the variance of normalized stretch ratios and therefore reflects the consistency of distortion across pairs (ideal=0). Together, the stress and distortion metrics distinguish cases where emotional spaces differ only by global rescaling from cases exhibiting more heterogeneous or anisotropic deformation. Only average distortion is reported in the main text; the remaining metrics appear in the appendix.

**Layer-Level Distortion Analysis.** Because distortion can vary substantially across depth, especially in transformer models, we compute distortion metrics at every sublayer and quantify the percentage of layers that exhibit high distortion for each dataset–model pair. Datasets exceeding a fixed distortion threshold (set from the distribution of values across all models and datasets) are marked as "high-distortion." Reporting the fraction of affected layers provides a more sensitive measure of representational fragility than averaging distortion over depth, which can obscure localized failure points.

**Effect of Dimensionality Reduction.** When comparing emotional spaces after projecting into the 50D synthetic subspace, dimensionality reduction inevitably increases measured distortion. We therefore evaluate both the full-space and 50D cases to assess whether models with already coherent emotional structure maintain that structure under compression, and whether models with fragile geometry degrade further.

**Linear Probing.** We train linear classifiers on activations projected into the synthetic emotional subspace and evaluate them on human-written datasets. This tests whether emotional information remains linearly recoverable after projection, complementing the geometric alignment metrics with a measure of functional decodability.

**ML-AURA.** ML-AURA quantifies how selectively a neuron responds to a specific concept by framing each neuron as a threshold-based detector (Suau et al., 2024). For a labeled dataset $D$, each

neuron's output is summarized per example using the maximum activation across tokens. These scalar responses are then ranked and evaluated using the area under the precision-recall curve, comparing neuron output against the presence or absence of the target concept. Neurons with high AUC-PR are designated as "experts" for that concept.

In our adaptation, the concepts are emotion categories. We apply ML-AURA in a one-vs-all setup for each emotion, scoring each neuron by how well it distinguishes a target emotion from all others. For each layer, we compute the proportion of neurons whose one-vs-all AUROC exceeds 0.9, treating these as emotion-selective units.

# 4 EXPERIMENTS

Using the tools presented in Section 3, we provide evidence that emotional representations in LLMs are structurally universal. We show that emotions are encoded in similar geometric subspaces across datasets, languages, and writing styles. This and all subsequent sections focus on LLaMA 3.1; analogous results for Olmov2 and Ministral are provided in the appendix.

## 4.1 DATASETS

To extract emotional representations, the synthetic dataset of (Reichman et al., 2025) is used, in which neutral sentences are rewritten into multiple emotions so that emotional variation becomes the principal structured difference across samples. This synthetic corpus is used solely to obtain clean, maximally polarized affect directions; all downstream evaluations, alignments, and probes are performed exclusively on human-written datasets.

The universality of emotional representations in LLMs are evaluated using eight diverse datasets, each offering explicit categorical emotion labels; datasets restricted to polarity or star ratings were excluded as too coarse. The collection spans languages, modalities, and styles: Go-Emotions contains English Reddit comments (Demszky et al., 2020), CARER covers English tweets (Saravia et al., 2018), SemEval-2007 Task 14 focuses on English news headlines (Strapparava & Mihalcea, 2007), EmoEvent includes English and Spanish tweets (Plaza-del-Arco et al., 2020), Emotions in Drama consists of German plays from the eighteenth and nineteenth centuries (Dennerlein et al., 2023), Bhaav offers Hindi short stories (Kumar et al., 2019), MultiEmotions-It provides Italian YouTube and Facebook comments (Sprugnoli et al., 2020), and EmoTextToKids features French journalistic and encyclopedic texts written for children (Étienne et al., 2024). Appendix B analyzes the structural and stylistic contents of each of these datasets.

The chosen languages are those for which high-quality emotion datasets exist and which are officially supported by LLaMA 3.1, as specified in its technical report.

## 4.2 UNIVERSALITY ANALYSIS

Using the hidden-state representations, projections, and metrics defined in Section 3, we evaluate how consistently emotions are encoded across datasets. Cosine similarity operates on emotion centroids, the regression MSE is derived from a linear map fitted to all mean-pooled-level activations sharing the same emotion label, and the stress/distortion metrics operate on cross-dataset, same-emotion distance pairs defined in the synthetic and real latent spaces.

Table 1 shows that all models exhibit high centroid cosine similarity between real and synthetic emotion directions (0.83–0.93), indicating stable cross-dataset encoding of emotional categories. English datasets are only marginally higher than non-English ones, suggesting near-equivalent representational fidelity across languages. Instruction-tuned variants show higher cosine similarity and lower regression error than their base models, reflecting closer alignment to the synthetic manifold. MSE values are broadly comparable across languages, with differences well within variance. By contrast, spectral flatness and Frobenius norms remain broadly similar across models and do not distinguish base from instruct variants: LLaMA shows a minor increase with tuning, OLMo shows none or a decrease, and Ministral remains in the same range. This pattern suggests that tuning improves representational alignment rather than relying on larger or more isotropic transformations.

| Model | Language | Avg Cos Sim ↑ | Stress-2 ↓ | Avg Dist ↓ | Probe Acc. ↑ | Avg MSE ↓ | Avg Spectral Flatness | Avg Frob Norm |
|---|---|---|---|---|---|---|---|---|
| Llama-Base | English | 0.84 ± 0.13 | 0.15 ± 0.14 | 0.97 ± 0.22 | 0.47 ± 0.14 | 1.81 ± 1.99 | 2.04 ± 0.39 | 7.54 ± 1.82 |
| Llama-Base | Non-English | 0.84 ± 0.12 | 0.18 ± 0.15 | 0.96 ± 0.22 | 0.4 ± 0.11 | 1.81 ± 2.00 | 2.10 ± 0.41 | 7.66 ± 2.45 |
| Llama-Instruct | English | 0.93 ± 0.08 | 0.22 ± 0.11 | 0.78 ± 0.12 | 0.4 ± 0.06 | 0.93 ± 1.09 | 2.26 ± 0.41 | 8.70 ± 6.73 |
| Llama-Instruct | Non-English | 0.94 ± 0.05 | 0.22 ± 0.15 | 1.01 ± 0.15 | 0.45 ± 0.06 | 0.89 ± 1.05 | 2.30 ± 0.41 | 8.66 ± 6.05 |
| Olmov2-Base | English | 0.88 ± 0.13 | 0.59 ± 0.85 | 1.46 ± 0.63 | 0.42 ± 0.05 | 1.90 ± 5.40 | 2.19 ± 0.39 | 7.48 ± 0.41 |
| Olmov2-Base | Non-English | 0.83 ± 0.16 | 0.61 ± 1.5 | 1.35 ± 0.78 | 0.45 ± 0.05 | 1.86 ± 5.30 | 2.35 ± 0.39 | 8.45 ± 2.05 |
| Olmov2-Instruct | English | 0.90 ± 0.10 | 0.32 ± 0.32 | 47%* | 0.47 ± 0.06 | 1.03 ± 2.24 | 2.20 ± 0.37 | 7.60 ± 0.64 |
| Olmov2-Instruct | Non-English | 0.89 ± 0.09 | 0.43 ± 0.59 | 51%* | 0.45 ± 0.05 | 0.97 ± 2.11 | 2.32 ± 0.40 | 8.30 ± 1.61 |
| Ministral | English | 0.94 ± 0.06 | 0.21 ± 0.27 | 1.11 ± 0.17 | 0.39 ± 0.05 | 1.73 ± 2.29 | 2.17 ± 0.43 | 7.53 ± 0.73 |
| Ministral | Non-English | 0.93 ± 0.09 | 0.24 ± 0.29 | 1.18 ± 0.12 | 0.45 ± 0.05 | 1.69 ± 2.25 | 2.24 ± 0.43 | 7.50 ± 0.83 |

Table 1: Per-model stress, distortion, linear-alignment metrics, and probe accuracy for English and non-English datasets. Lower distortion reflects greater geometric consistency. Cells marked with * indicate high stress/distortion; in these cases the table reports the percentage of layers exhibiting high distortion rather than raw scores. A full per-dataset breakdown appears in Appendix E.

Across datasets, LLaMA-3.1-8B-Base exhibits the lowest stress-2 values, indicating the most coherent emotional geometry (Table 1). Appendix E shows the same comparison broken down per dataset and includes additional stress and distortion metrics. Non-English datasets tend to have slightly higher stress-2 than their English counterparts. Average distortion varies on a model-by-model basis as to whether the English or non-English datasets have a higher score. LLaMA-3.1-8B-Instruct and Ministral also maintain relatively low Stress-2 scores, though both show a modest increase relative to the base model. OLMo-v2 displays a markedly different pattern: the base version shows substantially higher stress than LLaMA or Ministral, and the instruct variant is higher still, suggesting a less unified emotional space. Projecting representations into the 50D synthetic subspace increases stress for all models (shown in Appendix E), but the effect remains small for models with coherent geometry and is most pronounced for OLMo, where elevated stress persists after projection.

The average distortion metric further differentiates the models. LLaMA-3.1-8B-Base remains closest to the expected range for both English and non-English datasets, whereas OLMov2-Base shows substantially larger deviations. Distortion also behaves differently across languages: some models compress English representations (e.g., LLaMA-Instruct), others expand non-English representations (e.g., Ministral), and the direction of deviation is not consistent across base/instruct variants. For most models these shifts are modest, with OLMov2 exhibiting the most pronounced deviations. Across models, instruction-tuned variants generally show higher distortion than their base counterparts, with the notable exception of LLaMA-3.1-Instruct on non-English datasets, where distortion is nearly ideal. LLaMA-3.1-8B-Instruct therefore under-compresses English while improving non-English distortion, Ministral shows mild expansion while remaining stable, and OLMov2-Base has the largest distortion of any model, with OLMo-v2-Instruct pushing this even further, nearly half of its layers becoming severely over-distorted.

In terms of stress, however, the instruct variant of OLMo exhibits markedly lower stress than its base model. Ministral's stress-2 score is lower still, aligning closely with LLaMA-Instruct. The comparatively low stress-2 of OLMo-v2-Instruct alongside its extreme distortion suggests that instruction tuning improves global directional alignment while simultaneously degrading local geometric coherence, indicating that OLMo's emotional manifold may be more fragile and easily warped or over-rotated by tuning. Appendix E provides dataset-level breakdowns and additional stress and distortion measures, with further contextualization in Appendix D.

Certain dataset–model combinations exhibit extremely high distortion across large portions of the network when evaluated relative to the synthetic emotional space. These patterns are detailed in Appendix E. Even in cases with elevated distortion, however, substantial subsets of layers continue to preserve a coherent and transferable emotional geometry.

Linear probes achieve above-chance accuracy when predicting the emotions of human-written text from projected activations (Table 1) . This indicates that emotional structure in LLMs is not only geometrically consistent but also linearly decodable across diverse domains, though probe performance varies with model family and the coherence of the underlying emotional space.

Comparing the alignment metrics in Table 1 with the geometric faithfulness and structural stability metrics underscores a central tension. The layer-averaged cosine similarities and regression errors suggest that all models align well with the synthetic emotional manifold, often with small differences across families. Yet the stress and distortion metrics reveal that, within the same models, relational

structure can still be substantially warped—sometimes across large fractions of layers. This discrepancy reflects the fact that centroidal and regression-based measures capture global alignment, whereas stress and distortion expose finer-grained deviations in how relative distances between emotions are preserved. Thus, high apparent alignment at the aggregate level can coexist with local irregularities in the geometry of emotional spaces; however, linear probing shows that these spaces can still usefully and predictably predict emotion, even if there is some distortion between the synthetic and human-written spaces. Taken together, these results indicate that the claims concern the directional organization of the emotional subspace rather than strict isometry, and that the local warping revealed by stress and distortion is expected in high-dimensional compression while remaining compatible with a functionally coherent and manipulable emotional manifold.

## 4.3 MODEL PSYCHOLOGY

Having established the external consistency of emotional geometry across datasets, we now turn inward, asking how these emotions are internally structured within the model, and what this reveals about the model's implicit psychological architecture.

The first perspective examined is neural encoding patterns using ML-AURA (Section 3). Results focus on Llama3.1-8B-Base, with replications across models in Appendix F. Across the six Ekman emotions, an average of 75% of neurons per layer exceed the selectivity threshold, with sadness (98%) and surprise (97%) most pervasive, and fear lower (48%). This reflects sparse specialization rather than weak separability, and, importantly, selectivity rather than activation magnitude: ML-AURA identifies neurons whose responses discriminate emotions, not neurons that merely fire strongly. Non-Ekman emotions—envy, neutral, excitement—also show strong separability, averaging 88%.

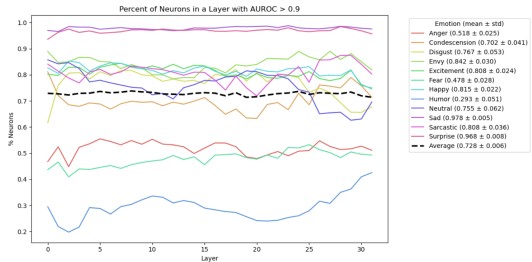

Figure 1: Results of ML-AURA by layer and emotion. Results are in terms of percent of neurons with an AUROC score above 0.9.

When analyzing by architectural component, MLP layers show slightly higher selectivity than attention layers (79% vs. 76.5%). Selectivity fluctuates across depth, with no clear monotonic trend: while the first layer starts at 76% and the final layer ends at 76.3%, several peaks and troughs occur in between, with the highest selectivity observed at layer 26 (79%). These patterns support the conclusion that emotional information is not confined to late layers or specialized regions, but is instead distributed broadly and redundantly throughout the network. These patterns are visualized by emotion in a layer-by-layer fashion in Figure 1.

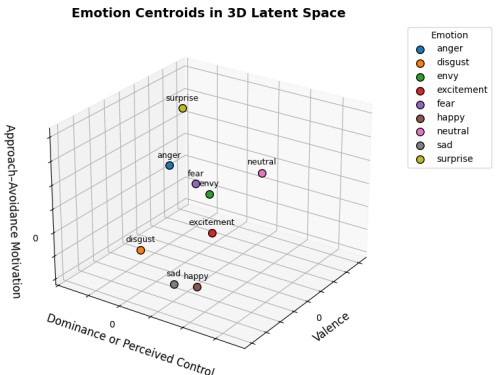

Figure 2: Emotion centroids plotted on the emotional axis found

To understand how emotions are geometrically represented in the network, we examine their structure within the derived SVD subspace. This subspace provides a low-dimensional lens into the model's internal affective organization. Our first goal is to assess how consistently emotions are arranged along the principal axes across layers and layer types.

We find that the emotional structure is remarkably stable across the Llama3.1-8B-Base model, particularly for the top three components. Across layers, the average Spearman correlation in emotion rankings is 0.87, 0.83, and 0.80 for PC1, PC2, and PC3, respectively; the corresponding Kendall's

Tau values are 0.82, 0.77, and 0.74. These results indicate that, while the magnitude and orientation of the components may shift, their semantic content remains intact.

Even when using a more fine-grained labeling scheme, as in the Go-Emotions dataset, which contains nearly three times as many emotion categories, we observe similar consistency. Rank-order correlations for Go-Emotions along the top three PCs remain high: Spearman values of 0.92, 0.74, and 0.73, and Kendall's Tau of 0.86, 0.68, and 0.68. These findings reinforce the conclusion that the model's emotional manifold is structurally stable, with interpretable axes. Appendix G reproduces the high consistency in how emotions are arranged along principal axes across all models studied.

Having established the stability of the SVD subspace across layers and datasets, the semantic content of the leading principal components is examined. Figure 2 visualizes the first three emotion axes that we describe below.

- PC1 strongly resembles a valence dimension. Emotions such as happy, surprise, and excitement lie at the positive end, while anger and fear occupy the negative end—suggesting a pleasure–displeasure continuum common to many psychological models.
- PC2 appears to reflect dominance or perceived control. Emotions high on this axis (e.g., fear, sadness) are often associated with low control or submission, whereas those at the opposite end (e.g., happy, surprise) may reflect more autonomous or socially detached states.
- PC3 maps onto approach–avoidance motivation. Emotions like excitement, happy, and envy—typically associated with goal-seeking behavior—score high, while anger and fear, more linked to avoidance or defensive responses, score low.
- PC4 may correspond to arousal or urgency. Surprise and fear rank highly, consistent with high physiological activation, while happy and neutral lie at the calmer end.

These dimensions are not explicitly supervised, but show surface-level resemblance to constructs proposed in affective science, such as valence, arousal, dominance, and approach–avoidance tendencies (cf. Russell (1980); Mehrabian (1996); Davidson (1995)). While these alignments are not exact, and many components blend multiple emotional signals, the emergence of such patterns suggests that large language models may implicitly encode affective distinctions that overlap with long-standing psychological taxonomies. This correspondence invites further investigation into the extent to which models trained solely on text internalize latent emotion structures, and whether these can serve as proxies or tools for understanding affective semantics in language.

Figure 3 provides a t-SNE visualization of the emotional space. Despite the dimensionality reduction, emotion classes form distinguishable, partially overlapping clusters, with closely related emotions (e.g., happy and excitement) frequently co-localized and others (e.g., fear and joy) appearing more spatially distant. While not all boundaries are sharp, the observed structure reinforces earlier findings: emotional information is embedded in a distributed yet semantically coherent geometry.

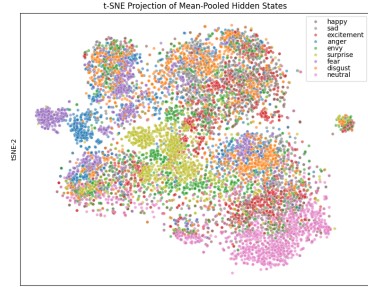

Together, the distributed AUROC patterns, stable subspace directions, interpretable principal components, and emergent clustering structure suggest that LLMs encode emotion not as isolated tags, but as coherent, multidimensional structures—akin to a learned affective manifold.

Figure 3: Plot of projected emotions in t-SNE space.

## 4.4 Steering Emotion Representations

Prior work on emotional steering in LLMs focuses primarily on shifting the emotional tone of generated text. (Subramani et al., 2022) and (Zou et al., 2023) learn vectors to modify output valence or categorical emotion. More recently, (Hollinsworth et al., 2024) attempts to steer internal emotional representations but collapses emotion into a binary positive/negative axis, achieving valence shifts 53.5% of the time. In contrast, we aim for fine-grained control over the model's internal perception of emotion across a full categorical space, while preserving semantic content.

We train a module that operates within the previously constructed SVD-based emotional subspace. For each emotion, we select all layers and sublayers where adding the centroid direction to same-

| Model | Language | Sad | Happy | Fear | Anger | Neutral | Disgust | Envy | Excitement | Surprise | Overall |
|-------|----------|-----|-------|------|-------|---------|---------|------|------------|----------|---------|
| **Llama3.1-8B** | English | 22 → 99 (0.14) | 20 → 92 (0.23) | 7 → 68 (0.25) | 21 → 58 (0.21) | 1 → 96 (0.21) | 0 → 88 (0.24) | 0 → 92 (0.24) | 0 → 90 (0.23) | 8 → 64 (0.22) | 9 → 83 (0.22) |
| **Llama3.1-8B** | Non-English | 9 → 100 (0.12) | 13 → 98 (0.21) | 5 → 68 (0.21) | 14 → 65 (0.20) | 16 → 95 (0.20) | 3 → 94 (0.21) | 0 → 99 (0.21) | 1 → 88 (0.21) | 7 → 64 (0.19) | **8 → 84 (0.20)** |
| **Olmov2** | English | 27 → 93 (0.09) | 32 → 98 (0.07) | 2 → 75 (0.05) | 21 → 100 (0.07) | 6 → 100 (0.04) | 1 → 99 (0.04) | 0 → 91 (0.04) | 0 → 81 (0.07) | 7 → 83 (0.07) | 11 → 91 (0.06) |
| **Olmov2** | Non-English | 33 → 84 (0.08) | 21 → 80 (0.07) | 2 → 59 (0.04) | 23 → 100 (0.06) | 8 → 99 (0.05) | 1 → 98 (0.04) | 0 → 84 (0.04) | 4 → 73 (0.08) | 4 → 91 (0.07) | 11 → 86 (0.06) |
| **Ministral** | English | 17 → 76 (0.09) | 24 → 90 (0.10) | 5 → 98 (0.09) | 17 → 100 (0.10) | 16 → 100 (0.10) | 2 → 96 (0.10) | 0 → 99 (0.10) | 1 → 82 (0.10) | 7 → 99 (0.10) | 10 → 91 (0.10) |
| **Ministral** | Non-English | 14 → 47 (0.09) | 28 → 60 (0.09) | 5 → 90 (0.08) | 22 → 97 (0.09) | 16 → 95 (0.09) | 3 → 87 (0.10) | 0 → 90 (0.10) | 3 → 51 (0.09) | 4 → 88 (0.09) | **11 → 79 (0.09)** |

Table 2: Top-1 prediction rates before and after learned steering for each target emotion across datasets and the semantic similarity loss between the hidden-state representations before and after steering. Each cell shows *baseline → post-steering (average semantic similarity loss)* accuracy.

emotion hidden states improves 1-vs-all classification AUROC beyond a fixed threshold. These layers are used for steering and serve as a proxy for the more challenging task of controllable emotional manipulation. At each selected layer, the model's hidden state is projected into the emotional latent space. The projected representation is passed through a one-layer MLP with a GELU activation to compute a shift, which is then mapped back into hidden-state space and added residually. The MLP is trained to steer the model's representation to favor the target emotion token when prompted.

We define the overall training objective as: $\mathcal{L}_{\text{total}} = \mathcal{L}_{\text{token}} + \mathcal{L}_{\text{sem}}$ where $\mathcal{L}_{\text{sem}}$ preserves semantic meaning and $\mathcal{L}_{\text{token}}$ enforces perceptual control.

**Semantic Preservation.** The semantic consistency loss combines cosine and $\ell_2$ distance between the original and shifted final-layer hidden states:

$$\mathcal{L}_{\text{sem}} = (1 - \cos(h_{\text{base}}, h_{\text{shifted}})) + \gamma \cdot \frac{\|h_{\text{base}} - h_{\text{shifted}}\|_2}{\|h_{\text{base}}\|_2 + \|h_{\text{shifted}}\|_2}$$

**Emotion Control.** To ensure accurate emotion classification, we combine a standard cross-entropy loss with a token-level margin loss. The margin loss enforces that the logit for the target emotion token $e_i$ exceeds its synonyms $s_i$ by a margin $m_1$ (0.5), and that both exceed all other emotions $e_j$ and their synonyms by $m_2$ (10):

$$\mathcal{L}_{\text{margin}} = \max(0, m_1 - (\log p_{e_i} - \log p_{s_i})) + \max(0, m_2 - (\log p_{s_i} - \log p_{e_j}))$$

To prevent the model from optimizing by suppressing unrelated tokens, we weight the loss for emotion tokens more heavily in $\mathcal{L}_{\text{CE}}$: $\mathcal{L}_{\text{token}} = \mathcal{L}_{\text{CE}} + \lambda \cdot \mathcal{L}_{\text{margin}}$

We optimize the objective using AdamW with learning rate 1e-3 and weight decay 1e-2, using a cosine schedule with 50 warm-up steps. Steering uses the top 40 dimensions of the centered SVD-derived emotional subspace. The learned module is trained independently for each target emotion across all selected steering layers, using supervision from emotion-token prompts and hidden-state consistency targets. At evaluation, token sampling is performed with temperature 0 for determinism.

The steering module is evaluated by measuring how reliably it shifts the model's emotion classification toward the target label while preserving semantic similarity. Table 2 shows that the learned steering approach achieves consistent and accurate control of internal emotional representations across English and non-English datasets for all three models. For most emotions, post-steering accuracy exceeds 80%, with many cases reaching 90–100%. Even for harder emotions and lower-resource languages, steering reliably improves target classification, with nearly all settings achieving at least 50% accuracy. Semantic similarity losses remain low across models and languages, indicating that steering preserves the core representational content while shifting affective interpretation. LLaMA-3.1-8B is the most steerable model across a plurality of settings, while OLMo-v2 shows the largest average improvement—despite OLMo-v2-Instruct previously exhibiting substantial geometric misalignment in Section 4. Appendix H reports full results per dataset, and Appendix I provides ablations isolating key methodological factors. Table 3 illustrates how emotional steering manifests in practice, with inputs rewritten to express the target emotion.

## 5 Conclusion

Using a combination of probing, alignment, and causal intervention techniques, this work shows that emotional representations in LLMs are directionally consistent across layers, datasets, and languages. We find that emotions cluster in coherent, low-dimensional subspaces whose structure is stable across

| Dataset | Original Text | Shift Emotion | Steering Rewrite |
|---|---|---|---|
| CARER (Twitter) | i waited in line longer than usual i didnt feel impatient that my business was delayed i listened to the master about why this was occurring and how i could be of service during that moment. | Anger | Are you kidding me?! I waited in line longer than usual and I'm still trying to contain my frustration that my business was delayed! I listened to the master about why this was occurring and how I could be of service during that moment, but honestly, it's just not good enough! |
| GoEmotions | Is that seriously the story? I haven't googled it. I just thought it was cool as a WW2 buff. | Happy | Is that seriously the most amazing story? I haven't googled it yet, but I just thought it was cool as a WW2 buff! |
| EmoTextToKids (FR) | Je savais que je prenais des risques | Fear | Je crains que je prenais des risques |

Table 3: Examples of text rewritten using the steering method.

architectural depth and transferable across linguistic and cultural domains. The leading axes of this space correspond to psychologically interpretable dimensions, despite no explicit supervision. These emotional directions are not confined to isolated neurons or layers but are distributed and redundant, supporting high linear separability even under one-vs-all probing. Alignment experiments further reveal that the synthetic and real-world emotion spaces can be matched with minimal distortion, and linear probes trained in one domain generalize well to others. Together, these findings suggest that LLMs internalize a structured latent affective manifold during pretraining.

Crucially, this representational structure is not merely interpretable but also controllable. Our learned intervention module achieves accurate and emotion-specific steering across languages, reliably shifting the model's internal affective state toward the desired target. Steering is especially effective for basic emotions like sadness, anger, and fear, even in low-resource settings. However, control over more nuanced categories such as envy and excitement remains inconsistent, particularly in Hindi, highlighting the residual challenges of lexical sparsity and affective ambiguity.

These findings offer a structured account of how LLMs represent and modulate emotion. Future work should extend this analysis to multimodal models, investigating whether shared affective subspaces emerge across language, vision, and speech, and whether emotional representations in one modality can steer or constrain perception in another. Such models may yield a richer, more disentangled affective geometry, enabling both deeper interpretability and more naturalistic emotional reasoning. Another interesting future direction is to examine the developmental trajectory of emotional representations during pretraining, although doing so would require access to intermediate training checkpoints of large-scale models.

**Limitations.** Our universality claim is conditioned on the language and style being reasonably well represented in the model's pretraining data. In out-of-distribution settings, such as nineteenth-century German drama or low-resource Hindi, the emotional latent space shows somewhat higher distortion and stress, yet remains directionally coherent, with probe accuracy above chance and successful steering. These results indicate that even under limited pretraining coverage, the learned emotional geometry retains usable structure rather than collapsing. Importantly, the steering experiments in Appendix H show that usable control remains achievable even in these regimes: Hindi, the weakest-performing language, still supports an absolute steering shift of roughly 50% on average across models, while German theater texts exhibit average shifts exceeding 80% across models. These results indicate that, although representational fidelity degrades when pretraining coverage is sparse, the underlying emotional manifold does not collapse; instead, it retains sufficient structure to support both detection and targeted intervention.

**Ethics Statement.** Because our method enables controlled shifts in a model's internal affective representation, we acknowledge the need to articulate its appropriate use and its limitations. The steering mechanism is intentionally constrained: it acts only on intermediate hidden states, preserves semantic content through an explicit alignment regularizer, and cannot induce or amplify harmful content beyond what the base model already permits. Nonetheless, certain emotions, especially high-arousal or interpersonal ones such as anger or contempt, may yield more forceful stylistic rewrites, reflecting asymmetries in how models encode these emotions. We therefore recommend applying steering only in transparent, user-directed contexts, such as tone adjustment, therapeutic or reflective writing tools, accessibility interfaces, or cross-emotion normalization for evaluation. The method is not intended for covert style manipulation, persuasion, or emotionally charged rewriting without user consent. Finally, variability in steerability across languages and domains (e.g., low-resource or archaic corpora) functions as an inherent boundary: the technique does not uniformly override model behavior but respects representational limits.

ACKNOWLEDGMENT

This work was supported in part by CoCoSys, one of the seven centers sponsored by the Semiconductor Research Corporation (SRC) and DARPA under the Joint University Microelectronics Program 2.0 (JUMP 2.0).

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

## A  APPENDIX

## B  STYLISTIC VARIATIONS AMONG SELECTED DATASETS

| Dataset | Sent Len | Syll/Word | Word Len | Dale-Chall | FK Grade | Sent Count | Sent Len Std | TTR |
|---|---|---|---|---|---|---|---|---|
| Synthetic-Emotions | 25.57 ± 14.31 | 1.58 ± 0.19 | 4.79 ± 0.69 | 10.82 ± 1.87 | 13.17 ± 6.13 | 2.09 ± 1.74 | 3.64 ± 5.37 | 0.864 ± 0.087 |
| Go-Emotions | 8.80 ± 5.26 | 1.38 ± 0.31 | 4.17 ± 0.83 | 8.26 ± 3.28 | 4.51 ± 4.07 | 1.58 ± 0.74 | 1.30 ± 2.03 | 0.960 ± 0.067 |
| CARER (Twitter) | 18.34 ± 10.68 | 1.40 ± 0.20 | 4.11 ± 0.57 | 8.50 ± 2.22 | 8.53 ± 4.75 | 1.10 ± 0.35 | 0.46 ± 2.07 | 0.908 ± 0.081 |
| SemEval | 6.34 ± 1.88 | 1.69 ± 0.36 | 5.22 ± 0.98 | 12.74 ± 3.40 | 6.84 ± 4.17 | 1.02 ± 0.14 | 0.03 ± 0.32 | 0.995 ± 0.031 |
| EmoEvents (EN) | 10.57 ± 6.81 | 1.64 ± 0.27 | 5.32 ± 0.96 | 10.77 ± 2.29 | 9.71 ± 4.14 | 2.57 ± 1.30 | 4.19 ± 4.00 | 0.930 ± 0.074 |
| EmoEvents (ES) | 11.41 ± 7.64 | – | 5.09 ± 0.84 | – | – | 2.38 ± 1.23 | 4.85 ± 4.65 | 0.920 ± 0.076 |
| German Drama | 8.50 ± 6.15 | – | 4.91 ± 0.96 | – | – | 2.08 ± 1.90 | 1.81 ± 2.94 | 0.960 ± 0.069 |
| MultiEmotions-It | 9.26 ± 7.93 | – | 5.15 ± 1.91 | – | – | 2.13 ± 2.03 | 2.55 ± 4.23 | 0.949 ± 0.113 |
| EmoTextToKids (FR) | 16.23 ± 9.55 | – | 4.73 ± 0.78 | – | – | 1.21 ± 0.55 | 0.82 ± 2.49 | 0.943 ± 0.064 |
| GSM-8K | 12.80 ± 4.78 | 1.34 ± 0.14 | 4.20 ± 0.39 | 9.52 ± 1.39 | 5.19 ± 2.52 | 3.43 ± 1.20 | 4.09 ± 2.75 | 0.705 ± 0.109 |

Table 4: Table of lexical and syntactic features per dataset. Dashes (–) indicate features were not computed due to language-specific constraints.

To evaluate the universality of emotional representations in LLMs, a diverse set of emotion classification datasets were selected spanning multiple languages, modalities, and writing styles. Only datasets with explicit categorical emotion labels were included; datasets with only polarity (e.g., positive/negative) or star ratings were excluded due to insufficient granularity. In total, eight datasets were selected:

1. Go-Emotions (Demszky et al., 2020): English Reddit comments
2. CARER (Saravia et al., 2018): English tweets
3. SemEval-2007 Task 14 (Strapparava & Mihalcea, 2007): English news headlines
4. EmoEvent (Plaza-del-Arco et al., 2020): English and Spanish tweets
5. Emotions in Drama (Dennerlein et al., 2023): German plays from the 18th–19th century
6. Bhaav (Kumar et al., 2019): Hindi short stories
7. MultiEmotions-It (Sprugnoli et al., 2020): Italian YouTube and Facebook comments
8. EmoTextToKids (Étienne et al., 2024): French journalistic and encyclopedic text aimed at children

Additionally, the GSM-8K dataset (Cobbe et al., 2021) is used as a "negative relief" to provide context for the stress and distortion figures computed later on. This provides a reference point for interpreting stress and distortion scores in the absence of emotional alignment.

Table 4 summarizes the lexicographic and stylistic metrics for all selected datasets, except Bhaav, which is written in a low-resource language with limited tooling support for extracting such features. The table shows that the datasets contain a great diversity in style and complexity. The synthetic dataset has the longest sentences, highest syllables per word, and highest Flesch–Kincaid (FK) Grade Level (Kincaid et al., 1975) score, making it among the most complex datasets to read. It is also the second most complex dataset in terms of the Dale–Chall readability score, which accounts for both average sentence length and the percentage of difficult words not on a list of familiar vocabulary (Dale & Chall, 1948). However, its type-token ratio (TTR) is the second lowest of all datasets, suggesting that despite its syntactic complexity, the vocabulary used is relatively constrained. Its intra-passage

sentence length variability is relatively high, as indicated by a high sentence length standard deviation (sent len std), reflecting a mix of short and long constructions within the same passage.

By contrast, the SemEval headlines dataset exhibits the shortest average sentence length and lowest sentence count, reflecting its highly compressed format. It nonetheless has the highest average word length and one of the highest syllables-per-word scores, indicating dense, information-packed language. Its extremely high TTR is likely inflated by its brevity, though it still reflects a wide lexical range given the short passage lengths. SemEval also has the lowest sentence length variability, with nearly zero variability, suggesting uniform sentence structure and a rigid rhetorical format.

The Go-Emotions dataset has the third-shortest average sentence length after SemEval and German Drama. It exhibits high TTR and mid-length passages, consistent with its source in colloquial online interactions. The short syntax paired with varied vocabulary reflects emotional expressiveness in informal registers. Its relatively low sentence length deviations suggests consistent sentence lengths across each example, reinforcing the impression of concise, focused expression.

CARER (Twitter) contains the second-longest sentences among all datasets, after Synthetic. It also shows a high TTR and relatively high syllables per word. This suggests that, despite being informal and social, the tweets in this dataset are lexically rich and syntactically expansive, likely due to elaboration or rhetorical emphasis often seen in emotional expression on social media. At the same time, the sentence length variability within each passage is low, indicating that tweets tend to follow a single syntactic rhythm rather than mixing short and long sentences.

The EmoTextToKids dataset, composed of journalistic and encyclopedic texts aimed at children, has the third-longest average sentence length across all datasets. Despite this, it only ranks in the middle for word length and lexical complexity. The moderately high TTR suggests deliberate lexical variation for educational purposes, balanced with readability suitable for younger readers. Its relatively low sentence length variability indicates syntactic regularity across sentences in each passage, appropriate for writing aimed at supporting comprehension.

The EmoEvents datasets, composed of Spanish and English tweets, occupy the middle range in sentence length but are among the highest in word length and syllables per word. EmoEvents-English in particular shows one of the highest sentence counts per passage. Both variants exhibit relatively high TTR, reflecting lexical variety within concise, affect-rich tweet structures. These datasets balance syntactic brevity with expressive density. EmoEvents also displays some of the highest sentence length variabilities, indicating significant variation in sentence length within a single tweet thread or message, likely due to stylistic shifts between exposition and reaction.

The MultiEmotions-It dataset is similar in lexical complexity, with high word length and moderately high TTR, but diverges structurally: it has the lowest sentence count per passage of any dataset. This suggests a more compact emotional style, especially compared to the more elaborated narratives in EmoEvents. The relatively high sentence length variability within passages suggests that even though few sentences are used, they vary in complexity and length.

The German Drama dataset is notable for its short sentence lengths—the second shortest overall after SemEval, but relatively high word length and TTR. This is consistent with dialogue-driven, emotionally loaded dramatic text, where each utterance is brief but lexically rich and expressive. Its sentence count per passage is high, suggesting frequent speaker turns or short, segmented lines of dialogue. The low sentence length variability reinforces the sense of rhythmic, evenly paced dialogue characteristic of dramatic form.

Finally, the GSM8K dataset shows structured, formulaic writing with moderately long sentences and the highest sentence count per passage. It has the lowest TTR across all datasets, reflecting its constrained and repetitive vocabulary, which is typical for procedural and instructional text. Despite the repetitive vocabulary, its sentence length variability is high, suggesting alternating short prompts and longer explanatory steps typical of math problems written in natural language.

These datasets span a wide range of styles and complexities, reflecting the linguistic and cross-linguistic diversity of the corpora. By aligning the emotional manifold across such varied textual forms, ranging from mathematical instruction to dramatic dialogue, headlines to encyclopedic writing, it becomes clear that the manifold is not merely encoding textual style (which varies significantly and inconsistently), but is instead capturing the underlying emotional content of the text.

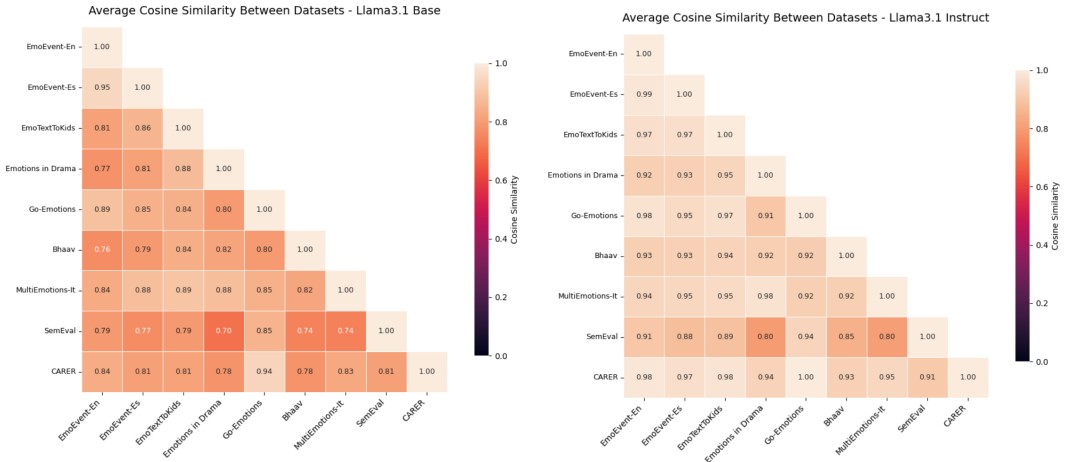

Figure 4: Cosine similarity of emotional centroids between datasets for Llama3.1-Base.

Figure 5: Cosine similarity of emotional centroids between datasets for Llama3.1-Instruct.

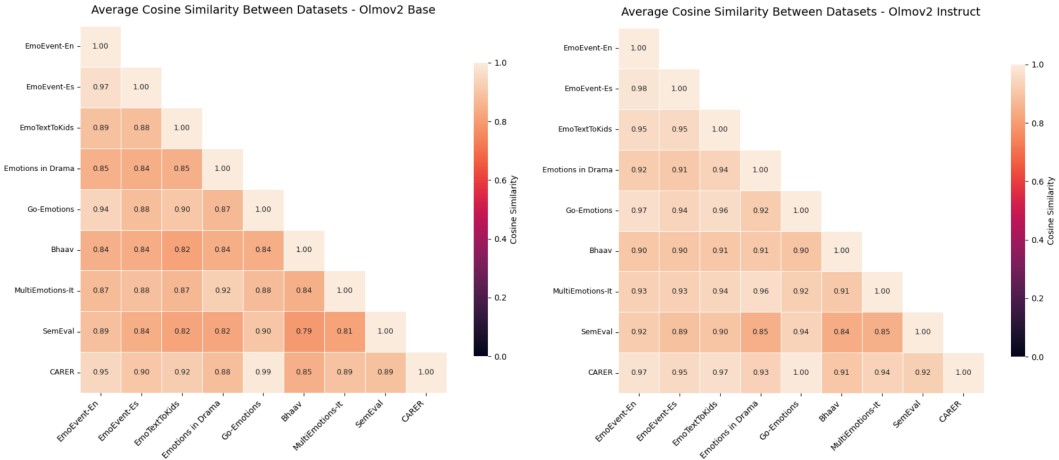

Figure 6: Cosine similarity of emotional centroids between datasets for Olmov2-Base.

Figure 7: Cosine similarity of emotional centroids between datasets for Olmov2-Instruct.

## C  COSINE SIMILARITIES BETWEEN DATASETS

Section 4 discussed the cosine similarity between emotional centroids in latent space across datasets. In this appendix, this is broken down by dataset across each model. Figure 4 presents the average cosine similarity between the synthetic dataset and the human-written dataset for Llama3.1-8B-Base. Figure 5 does so for Llama3.1-8B-Instruct. Figures 6 and 7 does so for Olmov2-8B-Base and Olmov2-8B-Instruct, respectively. Lastly, Figure 8 does so for the Ministral model. Throughout these figures, the average cosine similarity between the activations of the synthetically written emotion text and the human-written emotion text is quite high, showing how they are represented in an aligned fashion.

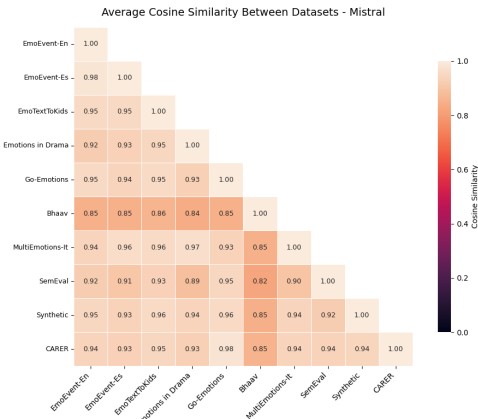

Figure 8: Cosine similarity of emotional centroids between datasets for Ministral.

## D    DISTORTION SCORE CONTEXTUALIZATION

To situate the stress and distortion scores reported in section 4, we tested the alignment between Llama3.1-8B-Instruct's synthetic emotion subspace and its activations on GSM8K. Although GSM8K is affect-neutral and task-oriented, "neutral" language is itself an emotional register; a truly emotionless baseline is impossible. Still, GSM8K provides a controlled case expected to align poorly with emotional structure. Consistent with this, it yields the largest distortions and among the highest stress values: Stress-1 = $0.517 \pm 0.130$, Stress-2 = $0.284 \pm 0.244$, Sammon stress = $22.84 \pm 32.41$, average distortion = $29.06 \pm 36.42$, L2 distortion = $134{,}869 \pm 168{,}270$, and sigma distortion = $14{,}435{,}586 \pm 11{,}306{,}312$.

These results confirm that alignment metrics only hold for datasets with consistent emotional structure: GSM8K, though emotionally neutral, is misaligned with the emotional manifold. Together with the lexical and syntactic diversity analysis presented in Table 4, these findings provide strong evidence that the observed structure in our aligned datasets is not an artifact of surface-level features, but instead reflects a consistent and abstract encoding of emotion in the model's internal geometry.

## E    UNIVERSALITY EXPANDED

In Section 4, stress-2 and average distortion scores for the models are discussed, with results compared across English and non-English datasets. In this appendix, the corresponding per-dataset results for stress, distortion, and probe accuracy are presented, along with an expanded set of stress and distortion metrics.

Table 5 presents the stress and distortion score per-model and per-dataset. LLaMA-3.1-8B-Base, which achieves the lowest stress values across datasets. LLaMA-3.1-8B-Instruct and Ministral also yield relatively low Stress-1 and Stress-2 scores, though somewhat higher than the LLaMA-3.1-Base model, suggesting that instruction-tuned variants preserve emotional relations slightly less consistently. Interestingly, OLMo-v2 shows the opposite trend: the base model achieves lower stress than the instruction-tuned variant, but both versions exhibit substantially higher stress values than LLaMA or Ministral. This pattern suggests that OLMo encodes a less unified emotional space than the other models. Projection into the 50D synthetic subspace increases stress across all models, consistent with expected compression effects. For models with already low stress values, the increases remain modest and the geometry remains coherent, whereas for models with elevated stress (e.g., OLMo), dimensionality reduction does not mitigate distortion.

To complement the stress scores, three high-dimensional embedding distortion metrics are reported: average distortion, $\ell_2$-distortion, and $\sigma$-distortion. These metrics highlight the contrast between base and instruction-tuned models. Both LLaMA-3.1-8B-Base and OLMo-v2-Base achieve distortion scores near ideal across most datasets (recall that no Ministral-Base model exists). For LLaMA-3.1-8B-Base, only two datasets deviate substantially from the ideal; excluding these outliers, its distortion

| Dataset | Stress-1 ↓ | Stress-2 ↓ | Sammon ↓ | Avg Dist ↓ | $\ell_2$ ↓ | $\sigma$ ↓ | Probe Acc. ↑ |
|---|---|---|---|---|---|---|---|
| **Llama3.1-8B-Base** | | | | | | | |
| Go-Emotions | 0.33 ± 0.33 | 0.13 ± 0.13 | | | 43%* | | 0.52 ± 0.52 |
| CARER (Twitter) | 0.38 ± 0.16 | 0.17 ± 0.15 | 0.18 ± 0.22 | 1.03 ± 0.24 | 1.11 ± 0.32 | 0.16 ± 0.33 | 0.35 ± 0.11 |
| SemEval | 0.34 ± 0.16 | 0.14 ± 0.15 | 0.14 ± 0.19 | 0.97 ± 0.23 | 1.03 ± 0.31 | 0.12 ± 0.26 | 0.46 ± 0.13 |
| EmoEvent (EN) | 0.36 ± 0.13 | 0.15 ± 0.13 | 0.15 ± 0.2 | 0.9 ± 0.19 | 0.97 ± 0.29 | 0.14 ± 0.26 | 0.55 ± 0.12 |
| EmoEvent (ES) | 0.38 ± 0.14 | 0.17 ± 0.14 | 0.16 ± 0.22 | 0.86 ± 0.2 | 0.93 ± 0.31 | 0.16 ± 0.29 | 0.5 ± 0.15 |
| Bhaav (Hindi) | 0.39 ± 0.15 | 0.17 ± 0.15 | 0.17 ± 0.19 | 0.86 ± 0.22 | 0.92 ± 0.31 | 0.16 ± 0.31 | 0.36 ± 0.1 |
| German Drama | 0.39 ± 0.17 | 0.18 ± 0.18 | 0.22 ± 0.47 | 1.06 ± 0.22 | 1.17 ± 0.48 | 0.48 ± 4.27 | 0.29 ± 0.1 |
| MultiEmotions-It | 0.38 ± 0.16 | 0.17 ± 0.15 | 0.19 ± 0.2 | 1.11 ± 0.24 | 1.19 ± 0.28 | 0.18 ± 0.48 | 0.46 ± 0.12 |
| EmoTextToKids (FR) | 0.41 ± 0.14 | 0.19 ± 0.15 | 0.19 ± 0.19 | 0.92 ± 0.24 | 0.99 ± 0.32 | 0.18 ± 0.33 | 0.39 ± 0.1 |
| Average (Full-Space) | 0.38 ± 0.15 | 0.17 ± 0.15 | 0.17 ± 0.25 | 0.97 ± 0.24 | 1.04 ± 0.35 | 0.2 ± 1.55 | 0.42 ± 0.14 |
| Average (50D-Space) | 0.58 ± 0.13 | 0.35 ± 0.16 | 0.33 ± 0.24 | 0.6 ± 0.21 | 0.68 ± 0.36 | 0.3 ± 1.66 | 0.39 ± 0.11 |
| **Llama3.1-8B-Instruct** | | | | | | | |
| Go-Emotions | 0.41 ± 0.11 | 0.18 ± 0.17 | | | 71%* | | 0.42 ± 0.06 |
| CARER (Twitter) | 0.4 ± 0.11 | 0.17 ± 0.11 | | | 40%* | | 0.58 ± 0.07 |
| SemEval | 0.6 ± 0.05 | 0.36 ± 0.06 | 0.32 ± 0.07 | 0.52 ± 0.09 | 0.56 ± 0.2 | 0.15 ± 0.28 | 0.49 ± 0.09 |
| EmoEvent (EN) | 0.4 ± 0.1 | 0.17 ± 0.09 | 0.2 ± 0.22 | 1.04 ± 0.14 | 1.15 ± 0.28 | 0.2 ± 0.3 | 0.11 ± 0.01 |
| EmoEvent (ES) | 0.43 ± 0.1 | 0.19 ± 0.1 | 0.22 ± 0.23 | 1.0 ± 0.14 | 1.12 ± 0.3 | 0.24 ± 0.38 | 0.11 ± 0.01 |
| Bhaav (Hindi) | 0.43 ± 0.11 | 0.2 ± 0.11 | 0.23 ± 0.3 | 1.02 ± 0.17 | 1.14 ± 0.38 | 0.23 ± 0.26 | 0.55 ± 0.05 |
| German Drama | 0.51 ± 0.15 | 0.29 ± 0.29 | | | 71%* | | 0.43 ± 0.07 |
| MultiEmotions-It | 0.49 ± 0.12 | 0.25 ± 0.13 | | | 46%* | | 0.57 ± 0.07 |
| EmoTextToKids (FR) | 0.42 ± 0.1 | 0.18 ± 0.1 | 0.21 ± 0.22 | 1.02 ± 0.13 | 1.13 ± 0.3 | 0.22 ± 0.27 | 0.60 ± 0.08 |
| Average (Full-Space) | 0.46 ± 0.12 | 0.22 ± 0.12 | 0.24 ± 0.23 | 0.92 ± 0.24 | 1.02 ± 0.37 | 0.21 ± 0.3 | 0.43 ± 0.19 |
| Average (50D-Space) | 0.55 ± 0.11 | 0.31 ± 0.13 | 0.33 ± 0.23 | 0.83 ± 0.27 | 0.96 ± 0.42 | 0.33 ± 0.35 | 0.37 ± 0.16 |
| **Olmov2-Base** | | | | | | | |
| Go-Emotions | 0.71 ± 0.36 | 0.63 ± 0.98 | 1.39 ± 5.32 | 1.65 ± 0.72 | 1.94 ± 1.08 | 0.34 ± 0.25 | 0.48 ± 0.06 |
| CARER (Twitter) | 0.66 ± 0.36 | 0.57 ± 1.03 | 1.22 ± 4.97 | 1.54 ± 0.67 | 1.8 ± 1.0 | 0.33 ± 0.25 | 0.61 ± 0.06 |
| SemEval | 0.78 ± 0.41 | 0.78 ± 0.95 | 1.35 ± 2.71 | 1.8 ± 0.63 | 2.02 ± 0.84 | 0.25 ± 0.18 | 0.6 ± 0.06 |
| EmoEvent (EN) | 0.56 ± 0.23 | 0.36 ± 0.42 | 0.75 ± 2.49 | 1.38 ± 0.5 | 1.61 ± 0.8 | 0.32 ± 0.24 | 0.11 ± 0.01 |
| EmoEvent (ES) | 0.55 ± 0.25 | 0.37 ± 0.5 | 0.76 ± 2.84 | 1.31 ± 0.59 | 1.53 ± 0.9 | 0.32 ± 0.22 | 0.11 ± 0.01 |
| Bhaav (Hindi) | 0.61 ± 0.26 | 0.44 ± 0.47 | 0.92 ± 2.86 | 1.26 ± 0.72 | 1.55 ± 1.09 | 0.44 ± 0.28 | 0.51 ± 0.05 |
| German Drama | 0.77 ± 0.58 | 0.93 ± 3.16 | 2.1 ± 12.92 | 1.42 ± 1.08 | 1.78 ± 1.53 | 0.52 ± 0.3 | 0.44 ± 0.05 |
| MultiEmotions-It | 0.74 ± 0.57 | 0.87 ± 2.74 | 1.88 ± 10.48 | 1.39 ± 1.04 | 1.71 ± 1.43 | 0.48 ± 0.28 | 0.58 ± 0.06 |
| EmoTextToKids (FR) | 0.6 ± 0.27 | 0.43 ± 0.59 | 0.87 ± 3.18 | 1.36 ± 0.47 | 1.62 ± 0.85 | 0.37 ± 0.33 | 0.59 ± 0.06 |
| Average (Full-Space) | 0.67 ± 0.4 | 0.6 ± 1.56 | 1.25 ± 6.43 | 1.46 ± 0.76 | 1.73 ± 1.1 | 0.37 ± 0.27 | 0.45 ± 0.2 |
| Average (50D-Space) | 0.72 ± 0.4 | 0.68 ± 1.61 | 1.56 ± 8.11 | 1.37 ± 0.9 | 1.78 ± 1.41 | 0.63 ± 0.42 | 0.34 ± 0.14 |
| **Olmov2-Instruct** | | | | | | | |
| Go-Emotions | 0.55 ± 0.19 | 0.34 ± 0.31 | | | 63%* | | 0.49 ± 0.06 |
| CARER (Twitter) | 0.57 ± 0.21 | 0.37 ± 0.41 | | | 38%* | | 0.65 ± 0.08 |
| SemEval | 0.46 ± 0.21 | 0.25 ± 0.3 | | | 47%* | | 0.61 ± 0.09 |
| EmoEvent (EN) | 0.51 ± 0.18 | 0.3 ± 0.26 | | | 41%* | | 0.11 ± 0.01 |
| EmoEvent (ES) | 0.51 ± 0.2 | 0.3 ± 0.3 | | | 41%* | | 0.11 ± 0.01 |
| Bhaav (Hindi) | 0.61 ± 0.27 | 0.44 ± 0.6 | | | 78%* | | 0.53 ± 0.05 |
| German Drama | 0.66 ± 0.29 | 0.52 ± 0.72 | | | 61%* | | 0.43 ± 0.05 |
| MultiEmotions-It | 0.65 ± 0.31 | 0.52 ± 0.88 | | | 41%* | | 0.56 ± 0.05 |
| EmoTextToKids (FR) | 0.56 ± 0.23 | 0.37 ± 0.45 | | | 34%* | | 0.62 ± 0.07 |
| Average (Full-Space) | 0.56 ± 0.24 | 0.38 ± 0.52 | | | 49%* | | 0.46 ± 0.20 |
| Average (50D-Space) | 0.65 ± 0.25 | 0.48 ± 0.58 | 979.03 ± 3086.26 | 882.19 ± 3055.12 | 482470.52 ± 978732.71 | 2673197.92 ± 3939472.37 | 0.36 ± 0.15 |
| **Ministral** | | | | | | | |
| Go-Emotions | 0.45 ± 0.17 | 0.23 ± 0.35 | | | 68%* | | 0.4 ± 0.05 |
| CARER (Twitter) | 0.47 ± 0.11 | 0.23 ± 0.11 | | | 70%* | | 0.49 ± 0.08 |
| SemEval | 0.38 ± 0.2 | 0.19 ± 0.51 | 0.24 ± 0.86 | 1.13 ± 0.19 | 1.22 ± 0.37 | 0.16 ± 0.49 | 0.55 ± 0.07 |
| EmoEvent (EN) | 0.41 ± 0.11 | 0.18 ± 0.1 | 0.21 ± 0.2 | 1.1 ± 0.14 | 1.2 ± 0.23 | 0.18 ± 0.15 | 0.11 ± 0.01 |
| EmoEvent (ES) | 0.48 ± 0.12 | 0.25 ± 0.12 | 0.32 ± 0.26 | 1.24 ± 0.14 | 1.37 ± 0.25 | 0.2 ± 0.16 | 0.11 ± 0.01 |
| Bhaav (Hindi) | 0.42 ± 0.11 | 0.19 ± 0.10 | | | 75%* | | 0.45 ± 0.05 |
| German Drama | 0.50 ± 0.25 | 0.32 ± 0.98 | | | 70%* | | 0.45 ± 0.07 |
| MultiEmotions-It | 0.49 ± 0.12 | 0.26 ± 0.14 | 0.31 ± 0.21 | 1.18 ± 0.11 | 1.31 ± 0.18 | 0.24 ± 0.26 | 0.66 ± 0.05 |
| EmoTextToKids (FR) | 0.43 ± 0.11 | 0.2 ± 0.12 | 0.24 ± 0.22 | 1.13 ± 0.12 | 1.24 ± 0.22 | 0.19 ± 0.16 | 0.57 ± 0.07 |
| Average (Full-Space) | 0.45 ± 0.16 | 0.23 ± 0.39 | 0.26 ± 0.44 | 1.16 ± 0.15 | 1.27 ± 0.27 | 0.19 ± 0.28 | 0.42 ± 0.19 |
| Average (50D-Space) | 0.52 ± 0.14 | 0.29 ± 0.4 | 0.32 ± 0.45 | 1.04 ± 0.18 | 1.19 ± 0.31 | 0.31 ± 0.3 | 0.36 ± 0.16 |

Table 5: Per-dataset distortion metrics and probe accuracy across three models. Lower distortion indicates greater geometric consistency. * in cells denotes very high stress/distortion. Instead of reporting the stress or distortion for that dataset, the percentage of layers that are highly distorted are reported.

metrics are more favorable than those of OLMo-v2-Base in both the full hidden-state space and the 50D synthetic subspace. See Appendix D for a contextualization of these scores.

The instruction-tuned models diverge considerably from their base counterparts. LLaMA-3.1-8B-Instruct produces roughly twice as many outlier datasets as LLaMA-3.1-8B-Base, while Ministral shows five outliers out of the nine datasets tested. Even after excluding these outliers, distortion remains higher for both models in both full-space and 50D-space analyses. Despite this, LLaMA-3.1-8B-Instruct and Ministral still maintain broadly consistent emotional structure across multiple datasets and languages. OLMov2-Instruct, however, performs markedly worse: the emotional spaces induced by different datasets are largely incompatible, with distortion elevated across nearly all layers and datasets.

Rather than reporting average distortion per layer for datasets with elevated distortion, we instead report the percentage of layers exhibiting high distortion. In LLaMA-3.1-8B-Base, only Go-Emotions

shows substantial distortion (43% of layers). The instruction-tuned variant shows broader distortion on Go-Emotions, though across datasets a majority of layers are affected only about half the time. OLMo-v2-Instruct shows majority distortion in 3 of 9 datasets, with the rest under 50% (some under 40%). Ministral is most fragile: when distortion appears, it usually spans most layers, though fewer datasets are affected. Together, these results indicate that even when a model exhibits distortion in parts of its architecture, large fractions of its layers still encode emotional spaces that remain universal across datasets, languages, and writing styles. These layers likely play a central role in maintaining consistent emotional representation within the model.

These per-dataset analyses show that while stress and distortion vary across model families and datasets, a broadly coherent emotional geometry is retained across datasets and languages, especially in the more stable base models. Probe accuracy trends reinforce this pattern, indicating that despite local geometric failures, large portions of each model preserve a shared, dataset-invariant emotional subspace.

## F    ML-AURA ACROSS MODELS

The ML-AURA analysis in Section 4.3 was reproduced with Llama3.1-8B-Instruct, Olmov2-Base OLMo et al. (2024) (11), Olmov2-Instruct (12), and Ministral-8B (AI, 2024) (13) showing results consist with what was found with Llama-3.1. Olmov2-Base performed worse than Olmov2-Instruct. Both of those models performed worse than Ministral-8B and the Llama at the same stage of training. The neurons in the Ministral-8B model were only slightly able than the neurons in the Llama model to separate between emotions. However, in all models more than the majority of neurons at all layers are able to do 1vAll classification of the specified emotion of interest at an AUROC > 0.9 showing great separability in how the different emotions are represented with a low amount of confusion.

Instruction-tuning was found to improve neuron's performance on separating emotion. For Llama-3.1-8B, instruction-tuning gave an average $4.8\%$ boost in the number of neurons per layer that were able to separate emotions.

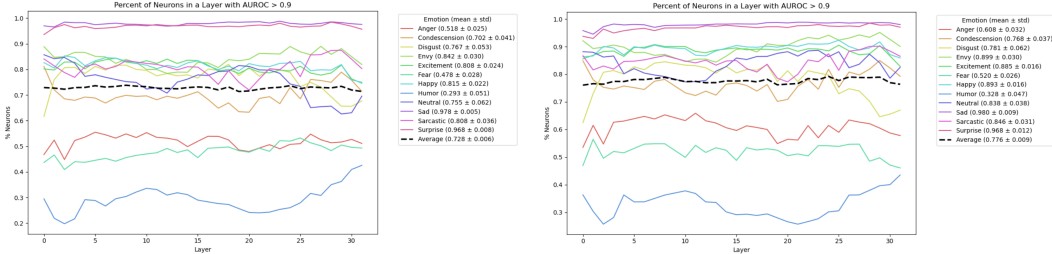

Figure 9: Results of ML-AURA by layer and emotion for LLama3.1-8B-Base (also in main paper, but reproduced here for ease of comparison with Llama3.1-8B-Instruct). Results are in terms of percent of neurons with an AUROC score above 0.9.

Figure 10: Results of ML-AURA by layer and emotion for LLama3.1-8B-Instruct. Results are in terms of percent of neurons with an AUROC score above 0.9.

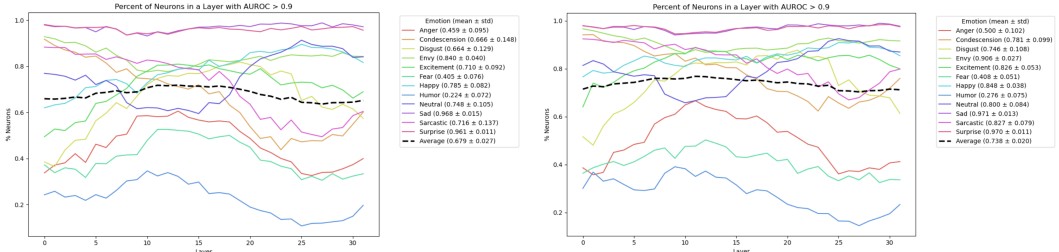

Figure 11: Results of ML-AURA by layer and emotion for Olmov2-Base. Results are in terms of percent of neurons with an AUROC score above 0.9.

Figure 12: Results of ML-AURA by layer and emotion for Olmov2-Instruct. Results are in terms of percent of neurons with an AUROC score above 0.9.

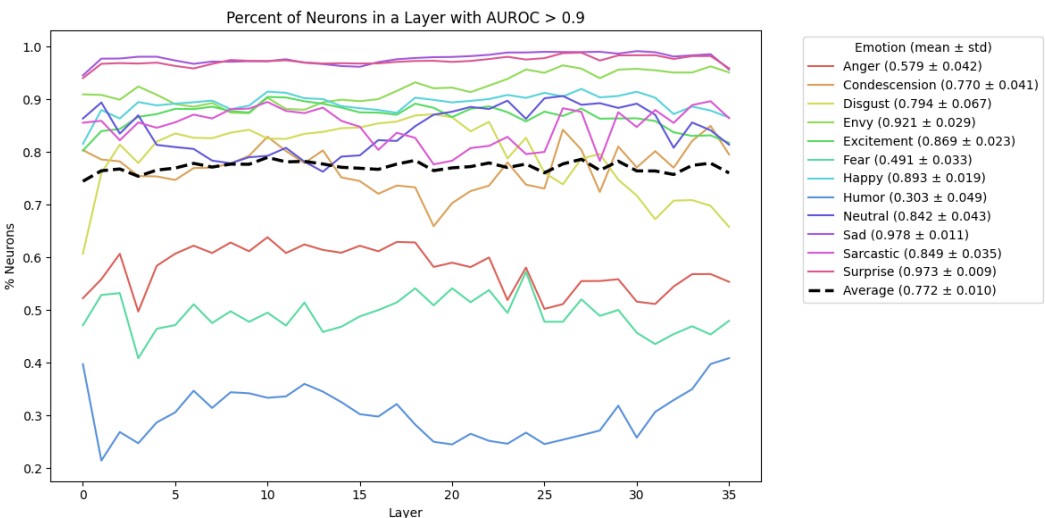

Figure 13: Results of ML-AURA by layer and emotion for Ministral. Results are in terms of percent of neurons with an AUROC score above 0.9.

## G    EMOTIONAL CONSISTENCY ACROSS PRINCIPAL AXES

| Model | PC | Avg. Spearman | Avg. Kendall |
|---|---|---|---|
| Llama3.1-8B-Base | PC1 | 0.87 | 0.82 |
| | PC2 | 0.83 | 0.77 |
| | PC3 | 0.80 | 0.74 |
| Llama3.1-8B-Instruct | PC1 | 0.87 | 0.81 |
| | PC2 | 0.92 | 0.85 |
| | PC3 | 0.84 | 0.78 |
| Olmov2-Base | PC1 | 0.76 | 0.71 |
| | PC2 | 0.70 | 0.65 |
| | PC3 | 0.69 | 0.64 |
| Olmov2-Instruct | PC1 | 0.84 | 0.77 |
| | PC2 | 0.74 | 0.69 |
| | PC3 | 0.70 | 0.65 |
| Ministral | PC1 | 0.94 | 0.91 |
| | PC2 | 0.83 | 0.83 |
| | PC3 | 0.79 | 0.79 |

(a) Synthetic emotion dataset.

| Model | PC | Avg. Spearman | Avg. Kendall |
|---|---|---|---|
| Llama3.1-8B-Base | PC1 | 0.92 | 0.86 |
| | PC2 | 0.74 | 0.68 |
| | PC3 | 0.73 | 0.68 |
| Llama3.1-8B-Instruct | PC1 | 0.98 | 0.98 |
| | PC2 | 0.85 | 0.78 |
| | PC3 | 0.70 | 0.65 |
| Olmov2-Base | PC1 | 0.76 | 0.70 |
| | PC2 | 0.69 | 0.65 |
| | PC3 | 0.68 | 0.64 |
| Olmov2-Instruct | PC1 | 0.83 | 0.77 |
| | PC2 | 0.74 | 0.69 |
| | PC3 | 0.70 | 0.65 |
| Ministral | PC1 | 0.93 | 0.90 |
| | PC2 | 0.74 | 0.68 |
| | PC3 | 0.71 | 0.66 |

(b) Go-emotions dataset.

Table 6: Correlation between emotion rankings in latent space across model layers.

| Model | PC | Avg. Spearman | Avg. Kendall |
|---|---|---|---|
| Llama3.1-8B-Instruct | PC1 | 0.73 | 0.68 |
| | PC2 | 0.82 | 0.76 |
| | PC3 | 0.81 | 0.75 |
| Olmov2-Base | PC1 | 0.75 | 0.69 |
| | PC2 | 0.86 | 0.79 |
| | PC3 | 0.77 | 0.72 |
| Olmov2-Instruct | PC1 | 0.74 | 0.68 |
| | PC2 | 0.78 | 0.72 |
| | PC3 | 0.77 | 0.71 |
| Ministral | PC1 | 0.76 | 0.69 |
| | PC2 | 0.84 | 0.77 |
| | PC3 | 0.79 | 0.71 |

Table 7: Correlation between emotion ranking in latent space across models. Each model latent space's emotion ranking in this table is being correlated to Llama3.1-8B-Base.

The emotional space is strikingly stable across models, especially along the first three principal components. Table 6 reports average Spearman and Kendall correlations of emotion orderings across layers for each model, showing consistently high values for both the synthetic dataset and the more fine-grained Go-Emotions labels. This suggests that the models' emotional manifolds possess stable, interpretable axes.

This structure is not only stable within models, but also consistent across them in terms of relative emotion positioning as shown in Table 7. The high rank-order correlations suggest that the emotional geometry described later in this section reflects a shared conceptual structure across models. However, results from the inter-model alignment analysis indicate that these shared structures are embedded in distinct internal coordinate systems, requiring high-complexity transformations to align. Thus, while the emotional manifolds are topologically consistent, their parameterizations remain model-specific—likely shaped by architectural and pretraining differences.

## H    EMOTIONAL STEERING PER DATASET RESULTS

In Section 4.4, the results of the steering method were presented as averages across English and non-English datasets. In this appendix, the corresponding per-dataset results are presented. Table 8 shows that the learned steering approach achieves consistent and accurate control over internal emotional representations across a diverse set of languages and datasets for all three models. For most emotions, post-steering accuracy typically exceeds 85% across models. Performance is robust even in multilingual settings, with particularly strong results in French, German, and Italian. Steerability remains high for many emotions in Hindi, a lower-resource language, suggesting that lexical sparsity

| Dataset | Sad | Happy | Fear | Anger | Neutral | Disgust | Envy | Excitement | Surprise | Overall |
|---|---|---|---|---|---|---|---|---|---|---|
| **Llama3.1-8B** | | | | | | | | | | |
| Semeval | 15 → 99 (0.15) | 23 → 91 (0.24) | 8 → 97 (0.26) | 23 → 59 (0.22) | 0 → 99 (0.22) | 0 → 97 (0.26) | 0 → 100 (0.26) | 0 → 96 (0.25) | 19 → 87 (0.23) | 10 → 92 (0.23) |
| CARER (Twitter) | 46 → 99 (0.13) | 16 → 88 (0.22) | 7 → 89 (0.23) | 10 → 43 (0.20) | 0 → 99 (0.20) | 0 → 95 (0.23) | 0 → 100 (0.23) | 0 → 85 (0.22) | 8 → 78 (0.20) | **10 → 86 (0.21)** |
| EmoTextToKids (FR) | 0 → 100 (0.13) | 5 → 97 (0.22) | 7 → 83 (0.23) | 11 → 64 (0.21) | 20 → 98 (0.21) | 6 → 96 (0.23) | 0 → 100 (0.23) | 0 → 96 (0.23) | 19 → 92 (0.20) | 8 → 92 (0.21) |
| German Drama | 10 → 100 (0.11) | 4 → 98 (0.19) | 9 → 51 (0.19) | 9 → 72 (0.18) | 0 → 98 (0.17) | 0 → 98 (0.19) | 0 → 95 (0.19) | 0 → 94 (0.20) | 10 → 73 (0.17) | 5 → 86 (0.18) |
| EmoEvents (EN) | 24 → 99 (0.14) | 23 → 97 (0.23) | 3 → 79 (0.23) | 41 → 79 (0.21) | 0 → 90 (0.21) | 0 → 92 (0.23) | 0 → 100 (0.23) | 0 → 95 (0.22) | 5 → 52 (0.21) | 11 → 87 (0.21) |
| EmoEvents (ES) | 19 → 99 (0.14) | 22 → 92 (0.23) | 9 → 92 (0.23) | 30 → 80 (0.21) | 0 → 89 (0.21) | 2 → 95 (0.23) | 0 → 100 (0.23) | 0 → 95 (0.23) | 3 → 60 (0.21) | 9 → 89 (0.21) |
| MultiEmotions-It | 9 → 99 (0.11) | 33 → 100 (0.20) | 0 → 82 (0.19) | 22 → 51 (0.18) | 6 → 92 (0.18) | 6 → 99 (0.19) | 0 → 100 (0.19) | 5 → 96 (0.19) | 4 → 73 (0.17) | **10 → 88 (0.18)** |
| Bhaav (Hindi) | 9 → 100 (0.13) | 0 → 51 (0.22) | 2 → 33 (0.23) | 0 → 59 (0.21) | 52 → 98 (0.21) | 0 → 83 (0.22) | 0 → 99 (0.23) | 0 → 58 (0.22) | 0 → 20* (0.19) | 7 → 67 (0.21) |
| GoEmotions | 3 → 99 (0.15) | 16 → 90 (0.23) | 8 → 8* (0.26) | 8 → 50 (0.21) | 4 → 97 (0.22) | 0 → 68 (0.25) | 0 → 69 (0.24) | 0 → 83 (0.24) | 0 → 40* (0.22) | 4 → 67 (0.22) |
| Overall | 15 → 99 (0.13) | 16 → 89 (0.22) | 6 → 68 (0.23) | 17 → 62 (0.20) | 9 → 96 (0.20) | 2 → 92 (0.23) | 0 → 96 (0.23) | 1 → 88 (0.22) | 8 → 64 (0.20) | 8 → 84 (0.21) |
| **Ministral** | | | | | | | | | | |
| Semeval | 8 → 73 (0.09) | 4 → 96 (0.09) | 6 → 98 (0.08) | 16 → 100 (0.08) | 53 → 100 (0.09) | 2 → 97 (0.10) | 0 → 99 (0.10) | 0 → 97 (0.09) | 8 → 99 (0.10) | **11 → 95 (0.09)** |
| CARER (Twitter) | 35 → 98 (0.09) | 26 → 98 (0.10) | 9 → 99 (0.08) | 15 → 100 (0.10) | 2 → 100 (0.09) | 0 → 99 (0.10) | 0 → 99 (0.10) | 0 → 83 (0.10) | 8 → 99 (0.10) | 11 → 97 (0.10) |
| EmoTextToKids (FR) | 14 → 66 (0.09) | 28 → 86 (0.09) | 12 → 99 (0.08) | 18 → 100 (0.10) | 11 → 100 (0.09) | 0 → 97 (0.10) | 0 → 100 (0.11) | 1 → 95 (0.09) | 13 → 100 (0.10) | 11 → 94 (0.09) |
| German Drama | 10 → 84 (0.09) | 28 → 43* (0.09) | 8 → 99 (0.08) | 16 → 100 (0.09) | 13 → 100 (0.09) | 4 → 99 (0.10) | 1 → 100 (0.10) | 2 → 92 (0.09) | 1 → 99 (0.09) | 9 → 91 (0.09) |
| EmoEvents (EN) | 14 → 82 (0.09) | 38 → 94 (0.10) | 2 → 98 (0.09) | 33 → 100 (0.10) | 2 → 100 (0.09) | 3 → 91 (0.10) | 0 → 99 (0.10) | 0 → 48 (0.10) | 7 → 97 (0.10) | **11 → 100 (0.10)** |
| EmoEvents (ES) | 23 → 20* (0.09) | 33 → 98 (0.10) | 2 → 97 (0.08) | 33 → 100 (0.10) | 1 → 97 (0.09) | 0 → 99 (0.11) | 0 → 22* (0.10) | 5 → 94 (0.10) | | 11 → 81 (0.10) |
| MultiEmotions-It | 7 → 34* (0.09) | 35 → 65 (0.09) | 1 → 98 (0.08) | 23 → 100 (0.10) | 8 → 98 (0.09) | 8 → 98 (0.10) | 0 → 99 (0.09) | 12 → 38* (0.09) | 1 → 94 (0.09) | 11 → 81 (0.09) |
| Bhaav (Hindi) | 14 → 30* (0.08) | 14 → 10* (0.08) | 1 → 55 (0.07) | 22 → 87 (0.08) | 46 → 82 (0.08) | 0 → 52 (0.08) | | 1 → 10* (0.07) | 1 → 50 (0.08) | 11 → 46 (0.08) |
| GoEmotions | 10 → 52 (0.10) | 28 → 73 (0.10) | 4 → 98 (0.09) | 4 → 100 (0.10) | 8 → 100 (0.11) | 2 → 95 (0.10) | 0 → 98 (0.11) | 3 → 98 (0.10) | 4 → 99 (0.10) | **7 → 90 (0.10)** |
| Overall | 15 → 60 (0.09) | 26 → 74 (0.09) | 5 → 93 (0.08) | 20 → 99 (0.10) | 16 → 97 (0.09) | 2 → 91 (0.10) | 0 → 94 (0.10) | 2 → 65 (0.09) | 5 → 92 (0.10) | 10 → 85 (0.09) |
| **Olmov2** | | | | | | | | | | |
| Semeval | 37 → 100 (0.10) | 30 → 100 (0.08) | 3 → 88 (0.04) | 9 → 100 (0.07) | 4 → 100 (0.04) | 1 → 100 (0.03) | 0 → 98 (0.04) | 0 → 92 (0.10) | 16 → 98 (0.07) | 11 → 97 (0.06) |
| CARER (Twitter) | 49 → 99 (0.09) | 26 → 100 (0.07) | 2 → 57 (0.05) | 6 → 100 (0.07) | 0 → 100 (0.04) | 0 → 99 (0.04) | 0 → 98 (0.04) | 0 → 70 (0.04) | 5 → 52 (0.08) | 10 → 86 (0.06) |
| EmoTextToKids (FR) | 42 → 100 (0.08) | 24 → 98 (0.07) | 4 → 86 (0.05) | 13 → 100 (0.07) | 0 → 100 (0.04) | 0 → 100 (0.04) | 0 → 98 (0.04) | 0 → 89 (0.09) | 15 → 99 (0.07) | **11 → 97 (0.06)** |
| German Drama | 44 → 87 (0.07) | 2 → 97 (0.06) | 5 → 76 (0.04) | 28 → 100 (0.07) | 0 → 100 (0.04) | 0 → 99 (0.04) | 0 → 94 (0.04) | 7 → 71 (0.08) | 0 → 98 (0.07) | 9 → 92 (0.06) |
| EmoEvents (EN) | 20 → 93 (0.08) | 49 → 96 (0.07) | 1 → 72 (0.04) | 26 → 100 (0.07) | 0 → 100 (0.04) | 1 → 99 (0.04) | 0 → 83 (0.04) | 0 → 79 (0.08) | 3 → 93 (0.07) | **11 → 91 (0.06)** |
| EmoEvents (ES) | 32 → 87 (0.08) | 40 → 86 (0.07) | 2 → 41* (0.04) | 23 → 100 (0.06) | 0 → 100 (0.04) | 1 → 100 (0.04) | 0 → 74 (0.04) | 0 → 85 (0.08) | 2 → 91 (0.07) | 11 → 85 (0.06) |
| MultiEmotions-It | 17 → 95 (0.08) | 38 → 91 (0.07) | 0 → 59 (0.05) | 28 → 100 (0.07) | 0 → 100 (0.07) | 2 → 99 (0.04) | 0 → 83 (0.04) | 14 → 77 (0.09) | 3 → 89 (0.07) | 11 → 88 (0.06) |
| Bhaav (Hindi) | 32 → 53 (0.08) | 0 → 28* (0.07) | 0 → 35* (0.04) | 25 → 98 (0.04) | 40 → 96 (0.05) | 0 → 94 (0.05) | 0 → 71 (0.04) | 0 → 45* (0.08) | 0 → 77 (0.07) | 11 → 66 (0.06) |
| GoEmotions | 3 → 78 (0.08) | 24 → 94 (0.07) | 1 → 78 (0.05) | 42 → 100 (0.07) | 20 → 100 (0.04) | 1 → 99 (0.04) | 0 → 86 (0.04) | 2 → 82 (0.08) | 3 → 88 (0.06) | 11 → 89 (0.06) |
| Overall | 31 → 88 (0.08) | 26 → 88 (0.07) | 2 → 66 (0.04) | 22 → 100 (0.07) | 7 → 100 (0.04) | 1 → 99 (0.04) | 0 → 88 (0.04) | 2 → 77 (0.09) | 5 → 87 (0.07) | **11 → 88 (0.06)** |

Table 8: Top-1 prediction rates before and after learned steering for each target emotion across datasets and the cosine similarity between the hidden-state representations before and after steering. Each cell shows *baseline → post-steering (average semantic similarity loss)* accuracy. *Indicates failure cases where target emotion remained under 10%.

and data imbalance remain limiting factors for certain emotions in under-resourced settings across models. The semantic similarity loss across all datasets, models, and emotions is low, indicating that steering preserves much of the original representational structure while enabling control of emotional perception. LLaMA-3.1-8B is the most steerable model for a plurality of emotions and datasets; however, Olmov2 shows the greatest average delta across emotions and datasets—an outcome notable given that Olmov2-Instruct was previously shown in Section 4 to struggle with representing emotions in a unified manner. Appendix I reports ablations isolating the method's key factors.

# I  ABLATIONS FOR EMOTIONAL STEERING

In Section 4.4, we introduced a method for steering how LLMs internally represent and perceive emotion. This appendix presents ablation studies identifying which components are essential for successful steering. We evaluate the impact of: (1) the number of steering dimensions in the SVD subspace, (2) the presence of the GELU nonlinearity, (3) the use of synonyms in the loss function, (4) the weight of the target-token term in the cross-entropy loss, (5) individual components of the semantic similarity loss, (6) the structure of the margin loss, and (7) the choice of target layers for intervention.

To reduce evaluation cost while capturing variance in performance, we selected three emotion-dataset pairs representing high, moderate, and poor performance in the main results: sad (EmoTextToKids), anger (CARER), and fear (Bhaav). All ablations were conducted using these fixed emotion-dataset combinations.

Table 9 presents the effect of varying the number of steering dimensions $R$ in the SVD subspace. We observe that extremely low ranks (e.g., $R = 1$) fail catastrophically, while small ranks like $R = 2$ surprisingly succeed on all three emotion-dataset pairs. However, this success is likely fragile—intermediate values such as $R = 15$ and $R = 10$ show inconsistent behavior, with performance collapses in some cases. As rank increases, steering generally improves, peaking around $R = 20$, which achieves near-perfect or perfect steering across all settings. Beyond this point, gains saturate or regress, particularly for fear, suggesting diminishing returns or overparameterization. We adopt $R = 20$ as the best-performing and most stable configuration.

Tables 10 and 11 examines the effect of varying the margin weights $m_1$ and $m_2$, which define separation constraints in the semantic loss. The margin $m_1$ enforces a minimum distance between the target emotion token and its synonyms, preventing collapse and encouraging meaningful local structure. We observe that performance remains relatively stable across $m_1$ values, though some

| Ablation Target | Sad (EmoTextToKids) | Anger (CARER) | Fear (Bhaav) |
|---|---|---|---|
| R=1 | $0.4 \rightarrow 0$ | $7.0 \rightarrow 100$ | $2.4 \rightarrow 0$ |
| R=2 | $0.4 \rightarrow 99.8$ | $7.0 \rightarrow 100$ | $2.4 \rightarrow 100$ |
| R=3 | $0.4 \rightarrow 37.9^*$ | $7.0 \rightarrow 100$ | $2.4 \rightarrow 100$ |
| R=5 | $0.4 \rightarrow 100$ | $7.0 \rightarrow 2.4^*$ | $2.4 \rightarrow 29.3^*$ |
| R=10 | $0.4 \rightarrow 64.3$ | $7.0 \rightarrow 99.6$ | $2.4 \rightarrow 44.2$ |
| R=15 | $0.4 \rightarrow 30.8$ | $7.0 \rightarrow 17.6^*$ | $2.4 \rightarrow 22.2^*$ |
| R=20 | $0.4 \rightarrow 93.2$ | $7.0 \rightarrow 99.1$ | $2.4 \rightarrow 81.3$ |
| R=25 | $0.4 \rightarrow 84.8$ | $7.0 \rightarrow 96.0$ | $2.4 \rightarrow 6.3^*$ |
| R=30 | $0.4 \rightarrow 85.4$ | $7.0 \rightarrow 68.4$ | $2.4 \rightarrow 65.2$ |
| R=35 | $0.4 \rightarrow 84.8$ | $7.0 \rightarrow 76.3$ | $2.4 \rightarrow 46.4$ |
| R=40 | $0.4 \rightarrow 99.7$ | $7.0 \rightarrow 42.7$ | $2.4 \rightarrow 32.7^*$ |
| R=45 | $0.4 \rightarrow 95.4$ | $7.0 \rightarrow 51.0$ | $2.4 \rightarrow 61.1$ |
| R=50 | $0.4 \rightarrow 99.2$ | $7.0 \rightarrow 99.3$ | $2.4 \rightarrow 27.2^*$ |
| R=100 | $0.4 \rightarrow 94.2$ | $7.0 \rightarrow 99.2$ | $2.4 \rightarrow 30.3^*$ |

Table 9: Ablation for number of steering directions. Top-1 prediction rates before and after steering under ablation conditions for selected emotion-dataset pairs. Each cell shows *baseline → post-ablation* accuracy. *Indicates failure cases where target emotion is not the most predicted Top-1 class.

| Ablation Target | Sad (EmoTextToKids) | Anger (CARER) | Fear (Bhaav) |
|---|---|---|---|
| m1=0.1 | $0.4 \rightarrow 99.2$ | $7.0 \rightarrow 66.7$ | $2.4 \rightarrow 37.3^*$ |
| m1=0.25 | $0.4 \rightarrow 97.8$ | $7.0 \rightarrow 99.0$ | $2.4 \rightarrow 27.1^*$ |
| m1=0.5 | $0.4 \rightarrow 99.7$ | $7.0 \rightarrow 42.7$ | $2.4 \rightarrow 32.7^*$ |
| m1=0.75 | $0.4 \rightarrow 96.1$ | $7.0 \rightarrow 99.8$ | $2.4 \rightarrow 22.2^*$ |
| m1=1 | $0.4 \rightarrow 93.3$ | $7.0 \rightarrow 65.4$ | $2.4 \rightarrow 37.25^*$ |

Table 10: Ablation for target synonym margin. Top-1 prediction rates before and after steering under ablation conditions for selected emotion-dataset pairs. Each cell shows *baseline → post-ablation* accuracy. *Indicates failure cases where target emotion is not the most predicted Top-1 class.

instability appears for *fear*, suggesting mild sensitivity. In contrast, $m_2$ enforces separation between the target emotion token and all other emotion tokens (and their synonyms). Steering is highly sensitive to this margin: low $m_2$ values consistently fail, while performance improves monotonically as $m_2$ increases. At $m_2 = 20$, all emotion-dataset pairs steer successfully, indicating that strong inter-class separation is essential. We adopt $m_1 = 0.75$, $m_2 = 20$ as the best-performing configuration.

Table 12 shows the effect of varying the weight of the cross-entropy loss applied to the target emotion token and its synonyms. Lower weights lead to poor steering, particularly on *fear*, while higher values generally improve performance. The best overall results are observed at a weight of 25, suggesting that strongly emphasizing the generation of target emotion tokens is necessary for effective control.

Table 13 reports ablations over discrete architectural and training choices. Removing the GELU activation severely degrades performance across all tasks, indicating that nonlinearity is critical for steering. Omitting bias has a moderate effect, while removing synonyms from the loss function leads to failure on *fear*, suggesting their inclusion helps generalize the steering signal. Within the semantic similarity loss, the delta-norm and cosine components can be individually removed with limited degradation, but removing the full loss results in collapse—suggesting a synergistic effect where both components reinforce each other to guide the model's representation. The emotion margin loss is also crucial—its removal results in failure across all settings. Finally, applying steering across all layers performs worse than selectively targeting layers based on alignment with the emotion direction, underscoring the importance of precise and informed intervention over blanket modification.

| Ablation Target | Sad (EmoTextToKids) | Anger (CARER) | Fear (Bhaav) |
|---|---|---|---|
| **m2=1** | 0.4 → 31.2 | 7.0 → 29.3 | 2.4 → 4.0* |
| **m2=2** | 0.4 → 51.9 | 7.0 → 99.0 | 2.4 → 3.4* |
| **m2=5** | 0.4 → 79.2 | 7.0 → 96.1 | 2.4 → 22.8* |
| **m2=10** | 0.4 → 99.7 | 7.0 → 42.7 | 2.4 → 32.7* |
| **m2=15** | 0.4 → 100 | 7.0 → 99.6 | 2.4 → 97.1 |
| **m2=20** | 0.4 → 99.6 | 7.0 → 100 | 2.4 → 100 |

Table 11: Ablation for margin between target and non-target classes. Top-1 prediction rates before and after steering under ablation conditions for selected emotion-dataset pairs. Each cell shows *baseline → post-ablation* accuracy. *Indicates failure cases where target emotion is not the most predicted Top-1 class.

| Ablation Target | Sad (EmoTextToKids) | Anger (CARER) | Fear (Bhaav) |
|---|---|---|---|
| **CE Loss Weight=1** | 0.4 → 96.3 | 7.0 → 95.1 | 2.4 → 1.4* |
| **CE Loss Weight=2** | 0.4 → 92.8 | 7.0 → 54.2 | 2.4 → 6.0* |
| **CE Loss Weight=5** | 0.4 → 94.9 | 7.0 → 98.7 | 2.4 → 12.7* |
| **CE Loss Weight=10** | 0.4 → 80.0 | 7.0 → 65.7 | 2.4 → 56.2 |
| **CE Loss Weight=15** | 0.4 → 89.8 | 7.0 → 85.2 | 2.4 → 56.7 |
| **CE Loss Weight=20** | 0.4 → 99.7 | 7.0 → 42.7 | 2.4 → 32.7* |
| **CE Loss Weight=25** | 0.4 → 98.0 | 7.0 → 99.8 | 2.4 → 93.2 |
| **CE Loss Weight=30** | 0.4 → 94.4 | 7.0 → 91.7 | 2.4 → 73.3 |

Table 12: Ablation for cross-entropy loss weight for emotion tokens. Top-1 prediction rates before and after steering under ablation conditions for selected emotion-dataset pairs. Each cell shows *baseline → post-ablation* accuracy. *Indicates failure cases where target emotion is not the most predicted Top-1 class.

| Ablation Target | Sad (EmoTextToKids) | Anger (CARER) | Fear (Bhaav) |
|---|---|---|---|
| **Baseline** | 0.4 → 99.7 | 7.0 → 42.7 | 2.4 → 32.7* |
| **No GELU** | 0.4 → 25.9* | 7.0 → 11.0* | 2.4 → 1.3* |
| **No Bias** | 0.4 → 88.2 | 7.0 → 91.7 | 2.4 → 26.9* |
| **No Synonyms** | 0.4 → 98.9 | 7.0 → 99.3 | 2.4 → 15.9* |
| **No Semantic Loss** | 0.4 → 30.2* | 7.0 → 88.9 | 2.4 → 100 |
| **No Cosine Loss** | 0.4 → 74.3 | 7.0 → 100 | 2.4 → 76.3 |
| **No Delta-Norm Loss** | 0.4 → 100 | 7.0 → 97.7 | 2.4 → 100 |
| **No Emotion Margin Loss** | 0.4 → 23.9 | 7.0 → 13.3* | 2.4 → 0.6* |
| **Target Layers=All** | 0.4 → 66.1 | 7.0 → 64.9 | 2.4 → 12.9* |

Table 13: Top-1 prediction rates before and after steering under various ablation conditions for selected emotion-dataset pairs. Each cell shows *baseline → post-ablation* accuracy. *Indicates failure cases where target emotion is not the most predicted Top-1 class.

