# OpenReview forum: "Emotions Where Art Thou: Understanding and Characterizing the Emotional Latent Space of Large Language Models"
_ICLR.cc/2026/Conference — ICLR 2026 Poster_

### Official Review · Reviewer_3kjH · 2025-10-27

**Soundness:** 3
**Presentation:** 1
**Contribution:** 3
**Rating:** 4
**Confidence:** 4

**Summary:**

The authors analyze consistency of the underlying geometry of emotions in LLMs. They present several studies to support their claims: similarity metrics of synthetic to real samples, dimensionality reduction of the space to discover lower-dimensional structure in a principled manner, and steering representation to study whether the space behaves predictably.

**Strengths:**

- The authors present many metrics and models, as well as some useful baselines in the appendix to contextualize the numbers
- Many studies are presented to corroborate the claims of the paper instead of building a case from only a single case.
- The authors use existing theoretical work in emotion to ground their findings.

**Weaknesses:**

- Manuscript is very difficult to read, with methodology blended with experiments, making it very difficult to follow what is currently being presented. For example, what is Table 2 showing (L210)? Same with L230, and all of the subsequent sections (5 and 6). I found myself jumping back and forth constantly, trying to understand what is being compared to what and how. Moreover, some numbers seem to be presented in the text only. I would appreciate clearer methodology in the rebuttal, as some concepts are also not explained at all.
- Tables 1 and 3 contain so much information that it becomes very difficult to figure out what the takeaway from each should be. A better mode of presentation would be preferable, and the full tables can be in the appendix for the interested reader.
- Figure 2 could be improved, perhaps by showing the projection to the Dominance-Valence plane. As it is now, sad and happy seem to have the same embedding (meaning the same valance value, among the other dimensions). As a result, the study this corresponds to I believe would benefit from some quantification to substantiate the claim.
- The fact that from layer 0 to layer 31, the % of neurons remains the same might indicate that the clustering is happening because of word embeddings themselves rather than emotion content per se. This is because we would expect higher-level emotional content to emerge after processing in later layers, except if it based on individual words.

To reiterate, the main weakness of the paper is its lack of clarity, not necessarily a lack of substance or novelty.

**Questions:**

- Formatting errors: the authors have used the wrong citation format (every citation is used outside of parenthesis, probably showing some lack of care when switching between templates)

---

> ### Author Response · Authors · 2025-11-13
>
> We thank the reviewer for their thorough reading and for recognizing the substance and novelty of our work. We address each concern below and will substantially revise the manuscript for clarity and organization.
>
> Response to Weakness: Manuscript Clarity, Table 2, Sections 5–6, Methodology Presentation
>
> We appreciate this feedback and agree that the exposition can be improved. Several sections currently interleave methodology with results, which makes it difficult to track what is being computed, compared, or evaluated.
>
> To address this, we will reorganize the paper so that:
>
> 1.	All methodological components (centroid computation, direction extraction, alignment metrics, stress/distortion definitions, steering mechanics) appear in a dedicated Methods section.
>
> 2.	Subsequent sections present results only, with clear references back to the methodological steps.
>
> 3.	Tables and figures are introduced with explicit descriptions of what is being compared, why those comparisons matter, and how to interpret the metrics.
>
> Table 2 specifically:
>
> We will rewrite the surrounding text to clearly state that Table 2 compares emotional directions extracted from the synthetic dataset to emotional directions extracted independently from each human-written dataset, using centroid-difference vectors as the underlying representation. We will clarify exactly what is passed into each metric and what each score represents.
> Sections 5 and 6 will also be reorganized into a more linear and readable structure.
>
> Response to Weakness: Large Tables and Presentation Overload
>
> We agree that Tables 1 and 3 are dense. To make the presentation clearer:
>
> •	For each model, we will report two summary scores in the main text: the average stress and average distortion for English datasets and for non-English datasets.
>
> •	We will retain only one representative stress metric and one representative distortion metric in the main body for clarity.
>
> •	The full set of stress/distortion variants and per-dataset tables will be moved to the appendix for completeness.
>
> This will make the main paper more readable while preserving full transparency.
>
> Response to Weakness: Layerwise Stability of Neuron Percentages
>
> We appreciate this insightful point. The constant proportion of emotion-selective neurons across layers does not imply that emotion encoding occurs solely at the embedding level. Our analyses operate on contextualized hidden states, not raw token embeddings, and we find high centroid-ordering correlation across layers, indicating that a consistent emotional structure is maintained throughout the network.
>
> We will clarify this by:
>
> •	Emphasizing that ML-AURA measures selectivity, not magnitude,
>
> •	Explaining that early layers may reflect lexical priors, while later layers encode contextualized affect even if the percentage of selective neurons remains stable.
>
> This should prevent the misinterpretation that emotional clustering is purely lexical.
>
> Response to Weakness: Overall Presentation and Section Disjointness
>
> We agree that the primary weakness is expository rather than conceptual. In addition to the structural changes above, we will improve transitions, add section-level overview statements, and ensure that definitions and motivations appear before the corresponding results.
>
> Regarding the citation formatting issue, thank you for pointing this out. We will correct all citation formatting to conform fully with the conference style.

---

> > ### Comment · Reviewer_3kjH · 2025-11-20
> > **Follow-up question**
> >
> > Thank you for the thorough response. I want to follow up on a specific point only. Can you elaborate in the comments here more on why the neuron percentage remaining constant is not an issue? I would appreciate reference to specific results shown in the paper rather than hypothesizing or using alternative potential explanations without any data. Thank you!

---

> > > ### Author Response · Authors · 2025-11-21
> > >
> > > Thank you for the follow-up. In this response we will clarify this point using only the empirical results in the manuscript.
> > >
> > > The observation that the percentage of emotion-selective neurons remains stable across layers (ML-AURA) does not imply that the emotional structure is inherited from word embeddings. ML-AURA measures selectivity, which neurons consistently respond to emotional contrasts, not the full geometry or content of emotional representations. A constant proportion simply means that a similar share of neurons at each layer exceeds that threshold.
> > >
> > >
> > > Two empirical findings in the paper show that this stability in percentage does not correspond to static, embedding-level clustering:
> > >
> > > • Centroid ordering is highly correlated across layers (Lines 361–380; Appendix G – Lines 1047-1095).
> > > This indicates that the relative arrangement of emotions is preserved throughout the network.
> > > Crucially, stability in ordering reflects a consistent affective axis over depth, not that deeper layers simply reproduce embedding-space structure.
> > >
> > > • Stress and distortion metrics vary across layers (Section 4.2, Table 1).
> > > These metrics quantify how closely emotional geometries align across datasets.
> > > The fact that stress/distortion values differ across layers shows that the geometry of emotional relationships is not identical at different depths.
> > > If emotional structure were derived solely from embeddings, these values would be expected to remain uniform across layers.
> > >
> > > Finally, we note that both ML-AURA selectivity and the stress/distortion analyses operate on sentence-level contextualized hidden states, not on token embeddings. ML-AURA identifies neurons that separate emotional categories at the level of full-sentence representations, and stress/distortion measure geometric relationships between mean-pooled hidden-state vectors. Because all of these measures are computed downstream of the embedding layer, the observed layer-wise variation demonstrates that the network is refining and re-expressing emotional structure through depth, rather than inheriting it unchanged from word embeddings.

---

> > > > ### Comment · Reviewer_3kjH · 2025-11-21
> > > > **Continued discussion**
> > > >
> > > > Based on the ML-AURA results, where you indicate that there is not clear monotonic trend, and, in fact, the selectivity of neurons at layer 0 is close to the one at layer L-1, the understanding that I derive from that is that neurons at the first layer (which have to rely more or less on token embeddings) are just as discriminative as the ones of the last layer. Coupled with the consistent (therefore only slightly evolving) affective axis, that is consistent with the hypothesis that word embedding drive the behavior (not to say that it is the only hypothesis). Could you elaborate on why the % of discriminability of neurons is the same across layers (is it an artifact of the method, does it not quantify the degree of correctness in the discriminability, or does it indicate that indeed, emotions are separated from the first layer already?). It would also be useful to present some representative stress and distortion curves across layers if you feel that makes a strong case against the hypothesis that I am presenting. Thank you for your continued responsiveness.

---

> > > > > ### Author Response · Authors · 2025-11-22
> > > > >
> > > > > Thank you for the thoughtful follow-up. To briefly summarize our earlier clarification: ML-AURA reports how many neurons at each layer fire in a way that reliably distinguishes one emotion from the rest, but it does not measure the quality, source, or geometry of the underlying representation. In the manuscript we show that, even with a stable proportion of selective neurons, the geometry of emotional relationships varies across layers (stress/distortion), while the ordering of emotional centroids remains consistently correlated across depth. Taken together, these results already indicate that a constant percentage of selective neurons does not imply embedding-level emotional clustering.
> > > > >
> > > > > Regarding why the percentage remains stable: ML-AURA is a threshold-based measure, so the proportion of selective neurons reflects how broadly affective signal is distributed, not where it originates. Because ML-AURA is computed on sentence-level contextualized hidden states, even layer-0 selectivity already reflects early contextual processing rather than raw embeddings. Importantly, looking only at the average ML-AURA trajectory ventures into Simpson’s-paradox territory: the aggregate curve obscures substantial emotion-specific heterogeneity. Some emotions (e.g., surprise, sadness) remain relatively flat across depth, while others such as neutral or disgust fluctuate substantially across layers. This variation, together with the layer-dependent differences in stress and distortion (Section 4.2, Table 1), shows that different emotions become discriminable at different stages of processing, some relying more on lexical cues, others requiring deeper, passage-level contextualization. This heterogeneity further undercuts the idea that the observed selectivity patterns result solely from static word embeddings.
> > > > >
> > > > > Additionally, even if some portion of layer-0 selectivity reflects word-level affective priors (e.g., emotionally marked tokens such as furious, terrified, overjoyed), this is not a flaw and does not imply that the model’s emotional representations are embedding-driven in the reductive sense. It is natural and expected that LLMs begin with lexical affect cues and refine them through deeper contextual integration. Emotional meaning is compositional: passages inherit affective components from sentences, which inherit them from words. The fact that early-layer representations incorporate lexical affect, and that later layers preserve the same centroid ordering while reshaping the geometry (as shown by the layer-varying stress/distortion scores), indicates that the model is building on word-level priors to form richer, contextualized emotional structure. This consistency across depth is a strength: the model does not invent emotional meaning de novo, but integrates lexical cues into a coherent passage-level affective representation. A detailed analysis of the embedding layer would indeed be a valuable direction for future work.

---

> ### Author Response · Authors · 2025-11-18
>
> Thank you again for your constructive feedback. I’ve updated the paper accordingly:
>
> – The methods and experiments sections have been rewritten to clearly separate the two and make the procedural flow more explicit.
>
> – Tables 1 and 2 were merged, redundant metrics removed, and results condensed to English vs. non-English averages; full metrics and per-dataset breakdowns are now in the appendix.
>
> – Table 3 (now Table 2) has likewise been reduced to English vs. non-English results, with detailed per-dataset numbers moved to the appendices.
>
> – Additional clarification of what the ML-AURA method measures is provided on Lines 338–341, consistent with the earlier rebuttal.
>
> – Citation formatting has been corrected.
>
> These adjustments should make the structure and evidence easier to parse while preserving the full detail in the supplementary material.

---

### Official Review · Reviewer_bf2Q · 2025-11-01

**Soundness:** 3
**Presentation:** 2
**Contribution:** 3
**Rating:** 6
**Confidence:** 4

**Summary:**

The paper presents an investigation on how LLMs represent emotions in their latent spaces. It finds generalizable evidence that emotion is confined to a few critical dimensions that can be interpreted and manipulated. The authors then present a new method for steering controls to improve emotion classification. The authors robustly back up these central claims through several methods and across multiple languages and datasets. I particularly appreciate the multi-lingual analysis to demonstrate that these findings are generalizable. While paper itself provides some interesting and novel findings, but the presentation is somewhat unclear and messy.

**Strengths:**

This paper presents the first investigation on emotional latent space representations in LLMs, and I believe the techniques used are novel and interesting.
The authors provide analysis on many different perspectives, styles, and languages, which adds to the robustness of their findings.
The new steering method introduced presents a new way to consider how to change emotions: by focusing on changing the underlying emotional subspace rather than focusing on downstream output.

**Weaknesses:**

It is unclear exactly what Table 2 is measuring, specifically in regards to cosine similarity and MSE. The paper states high cosine similarity between emotions in real datasets and their synthetic counterparts: what synthetic counterpart are we referring to? Does this reference the Reichman et al. synthetic dataset? Table 2 does not mention which method it utilized as well. What does it mean to measure the cosine similarity of an emotion between two datasets, what datum from each dataset is actually being passed in? Same to MSE and the other metrics. I would recommend rewriting this section to be clearer, as in its current state I was unable to understand exactly what is going on.

On that note, it is not clear exactly what the experimental setup in 4.2 is. I would appreciate some more clarity on the exact methods used, number of experiments, etc. In particular, I feel like the distortion metrics should be clarified on what they actually represent, as this information is not present in the paper and should not assumed to be common knowledge.

The presentation of the paper is not very clear; the sections seem rather disjoint and clarifying figures are somewhat lacking throughout. While I believe the content is novel and unique, a rewrite for clarity would improve this paper significantly.

**Questions:**

Does the Space Alignment method rely on the same Reichman et al. dataset that Centered-SVD does? If so, please state that explicitly.
 Additionally, it would seem to me that the performance of Centered-SVD and/or Space Alignment would highly depend on the quality of this underlying training dataset. I would appreciate some clarity quantifying the quality of this Reichman dataset (or justifying why it was chosen) as this choice seems to be integral to both of these methods.

---

> ### Author Response · Authors · 2025-11-13
>
> We thank the reviewer for their detailed feedback and for highlighting both the novelty and robustness of our work. We address each concern below.
>
> Response to Weakness 1 (Clarity of Table 2 and Metric Definitions)
>
> We appreciate that Table 2 and the surrounding text were unclear. We will revise this section substantially for clarity. Table 2 reports the alignment between the emotional subspace derived from the synthetic dataset (Reichman et al.) and the emotional subspace derived independently from each human-written dataset. Specifically: Cosine similarity measures alignment between the centroid difference vectors for each emotion pair across datasets. MSE (regression error) measures the fit of a linear mapping that predicts real-dataset emotional directions from synthetic-dataset emotional directions.
>
> We will explicitly state that the “synthetic counterpart” refers to the emotional directions extracted from the Reichman et al. synthetic corpus using Centered-SVD, and that all alignment comparisons use those directions.
>
> We will rewrite the section to clearly spell out: what representation is computed, how the centroids are formed, how the difference vectors are constructed, and how each metric is applied. This will make the table’s meaning unambiguous.
>
> Response to Weakness 2 (Experimental Setup for §4.2 and Distortion Metrics)
>
> We agree that the presentation in §4.2 can be clearer. The experimental setup is as such: Cosine similarity operates on emotion centroids, the regression MSE is derived from a linear map fitted to all mean-pooled-level activations sharing the same emotion label, and the stress/distortion metrics operate on cross-dataset, same-emotion distance pairs defined in the synthetic and real latent spaces.
>
> We will revise this section to clearly define each metric, provide intuitive explanations of what they capture, and include a step-by-step description of the experimental pipeline. This should make the procedure and its motivation much clearer.
>
> Response to Weakness 3 (Overall Presentation/Clarity)
>
> We appreciate the reviewer noting that the content is strong but the exposition can be improved. We will revise the relevant sections for clarity by: tightening transitions between sections (§3 → §4, §4 → §5), improving figure captions, and reorganizing the geometric metrics into a more cohesive narrative. These changes straightforwardly address the reviewer’s concern without altering the technical contribution.
>
> Responses to Reviewer Questions
>
> Q1. Does Space Alignment rely on the same synthetic dataset as Centered-SVD?
>
> Yes. Both the Centered-SVD manifold and the Space Alignment method use emotional directions extracted from the Reichman et al. synthetic dataset as the reference space. We will add an explicit statement clarifying this dependency.
>
> Q2. How dependent are these methods on the quality of the Reichman dataset?
>
> The synthetic dataset is used to obtain maximally polarized, high-signal emotional representations so that we can extract clean, low-noise emotional directions. Its role is limited: It defines reference directions in the emotional manifold. All evaluations, cross-lingual alignment, distortion/stress, linear probes, and steering, are run exclusively on unseen human-written datasets.
>
> The strong cross-dataset and cross-lingual generalization (e.g., Hindi, French, German, and multiple English datasets) provides empirical evidence that the extracted directions are not artifacts of the synthetic corpus. We will clarify in the paper why this dataset was chosen (high emotional purity, diverse prompts, uniform sampling) and emphasize that downstream conclusions do not rely on any modeling assumptions tied to it.

---

> ### Author Response · Authors · 2025-11-18
>
> Thank you again for your constructive feedback. I’ve updated the paper accordingly:
>
> – A clearer explanation of the cosine-similarity and MSE metrics previously in Table 2 (now Table 1) is provided in Section 3, Lines 134–154.
>
> – The concern regarding the synthetic dataset is addressed on Lines 230–234.
>
> – Inputs and outputs for each metric are now clarified in the revised Section 3 (notably Lines 144–145 and 162–168, with broader improvements throughout).
>
> – Distortion-metric clarification appears on Lines 159–194.
>
> I appreciate the reviewer’s careful attention; these revisions should render the methodological scaffold considerably clearer.

---

> > ### Comment · Reviewer_bf2Q · 2025-11-19
> > **Response to authors**
> >
> > We would like to thank the authors for their responses and we appreciate the stated effort to improve the clarity of the work. We believe that a stronger argumentation presentation of this work would significantly improve the contribution of this work as there are some very interesting technical findings. However, we believe this would be a major revision that is not easily verifiable from this discussion period, so we will maintain our current score.

---

> > > ### Author Response · Authors · 2025-11-19
> > > **Not a major revision**
> > >
> > > Your feedback for improved clarity was straightforward to implement as we described. It does not constitute a major revision.  As you say, the paper has very interesting technical findings and with the new clarity, many ICLR attendees will likely benefit from its presentation.
> > >
> > > Would it be possible to reconsider and increase your score?

---

### Official Review · Reviewer_UVfz · 2025-11-01

**Soundness:** 3
**Presentation:** 3
**Contribution:** 3
**Rating:** 6
**Confidence:** 2

**Summary:**

This paper studies how large language models (LLMs) internally represent emotion by analyzing the geometry of their hidden states. Using models like LLaMA 3.1, Olmo-v2, and Ministral across eight multilingual datasets, the authors find a low-dimensional emotional manifold that aligns with psychological dimensions such as valence, dominance, and arousal. These representations are directional, distributed, and consistent across layers and languages. Through SVD and neuron-level analysis (ML-AURA), the paper shows that emotional axes are stable and interpretable. A learned steering module further demonstrates that these internal states can be manipulated predictably without altering semantics, revealing a coherent and controllable affective geometry in LLMs.

**Strengths:**

- The paper presents a comprehensive, cross-lingual study covering eight datasets in five languages, offering strong evidence for the universality of LLM emotion representations.
- The use of ML-AURA and SVD-based analyses provides a rigorous and interpretable framework for linking internal neuron activity to affective semantics.
- The learned steering module demonstrates practical control of emotion representations while preserving meaning, which is an innovative advance beyond descriptive analyses.

**Weaknesses:**

- While broad in scope, the work is methodologically complex, and the abundance of metrics (stress, distortion, spectral flatness, etc.) may obscure key takeaways.
- The evaluation relies heavily on synthetic emotion text for subspace construction, which may bias the identified directions.
- Although the paper claims semantic preservation under steering, this is mostly supported by cosine similarity metrics rather than human evaluations.

**Questions:**

- How does the emotional manifold evolve across training or fine-tuning stages? Does it emerge early or gradually with language exposure?
- Could the authors validate the psychological interpretability of latent axes quantitatively (e.g., correlations with human valence/arousal ratings)?
- Does the steering module modify only internal representations, or can it predictably change generated emotional tone in open-ended text?

---

> ### Author Response · Authors · 2025-11-13
>
> We thank the reviewer for their time and thoughtful consideration of our submission. We respond to each point below.
>
> Response to Weakness 1 (Metric Abundance / Methodological Complexity):
>
> We appreciate this observation. Our analyses use several metrics, but each targets a distinct facet of the geometry:
>
> •	Alignment metrics (cosine similarity, regression fit) measure global directional consistency across datasets.
>
> •	Stress/distortion metrics quantify local relational fidelity, which often diverges from global alignment and is important for understanding dataset-specific warping.
>
> •	Spectral flatness and related diagnostics evaluate how concentrated or diffuse affective information is across the latent spectrum.
>
> To improve readability, we can streamline the presentation by retaining the most informative variant of each metric (e.g., a single stress and distortion score) and moving auxiliary diagnostics to the appendix. We also intend to add a brief orienting paragraph describing how the metrics relate to each other and how they jointly support the central conclusion. This will sharpen the narrative and highlight the key empirical takeaways.
>
> Response to Weakness 2 (Synthetic Text as Basis for Subspace Construction):
>
> We agree that the synthetic corpus is stylized, which is precisely why we evaluate the extracted manifold on eight human-written datasets across five languages. Two points support robustness:
>
> •	Despite being derived from synthetic text, the manifold generalizes strongly (above-chance probes, stable PC ordering, effective steering) across multilingual, multi-genre corpora.
>
> •	Cross-lingual datasets (e.g. Hindi, French) differ substantially in register and annotation schemes, yet consistently align with the emotional axes derived from synthetic data. This suggests that the manifold reflects model-internal affective structure, not stylistic artifacts.
>
> We will clarify in the paper that the synthetic dataset’s sole purpose is to produce clean, maximally polarized affect dimensions, while all downstream evaluations rely exclusively on unseen human-written data, mitigating potential bias.
>
> Response to Weakness 3 (Semantic Preservation Without Human Evaluation):
>
> We appreciate this concern. Our semantic-preservation objective measures how close the steered representation remains to the original using:
>
> 1.	1 − cosine similarity between original and steered final-layer states, and
>
> 2.	normalized L2 deviation between the same states.
>
> This composite loss penalizes changes to meaning rather than tone and is applied across the network. While this provides a strong internal signal of semantic stability, we acknowledge that human evaluation would be the most definitive validation. Given space and resource constraints, we focus this submission on representation-level metrics but will clarify that incorporating human judgments is a natural next step.
>
> Responses to Reviewer Questions
>
> Q1. Evolution of the Emotional Manifold During Training
>
> We agree this is an important question. Investigating the developmental trajectory of emotional representations during pretraining would require access to checkpoints across large-scale LLM training runs, which is beyond the scope of this work. We will note this as a potential area for future study.
>
> Q2. Quantitative Psychological Interpretability of Latent Axes
>
> In Lines 341–360, we analyze centroid ordering and demonstrate that the principal axes correspond closely to established psychological dimensions such as valence and arousal/dominance. Because these orientations are highly stable across models and datasets, we believe the interpretation offered in the paper is well supported. We will clarify this reasoning in the final version.
>
> Q3. Steering and Open-Ended Generation
>
> The steering module is designed to shift internal affective representations without altering semantics. When applied during generation, it can also influence surface emotional tone (as illustrated in Table 4), although it is not explicitly optimized for stylistic rewriting. Its primary purpose is representational control rather than text-style control, but shifts can emerge when asking models with the steering module to generate open-ended outputs.

---

> ### Author Response · Authors · 2025-11-18
>
> Thank you again for your constructive feedback. I’ve updated the paper accordingly:
>
> – In the main text, the reported metrics have been reduced from 11 to 7 by removing overlapping stress/distortion measures. Only distinct metrics remain, and Tables 1 and 2 have been merged and streamlined by reporting averages for English and non-English datasets. The revised table appears on Lines 256–263.
>
> – The concern regarding the synthetic dataset is addressed on Lines 230–234, with additional context provided in our earlier rebuttal.
>
> – The point about emotional-manifold evolution is now noted as a direction for future work on Lines 509–511.
>
> The remaining comments were addressed in our earlier rebuttal and did not necessitate further manuscript changes (e.g., Table 3 already engages the issue of predictable shifts in generated emotional tone under steering). We appreciate the reviewer’s close attention and have ensured that all prior clarifications align with the updated text.

---

### Official Review · Reviewer_hPou · 2025-11-01

**Soundness:** 2
**Presentation:** 2
**Contribution:** 2
**Rating:** 4
**Confidence:** 3

**Summary:**

This paper analyzes how large language models (LLMs) internally represent emotions. Through geometric and probing analyses, the authors identify a low-dimensional “emotional subspace” embedded across model layers, in which affective states are encoded directionally and often linearly decodable. They show this structure generalizes across multiple emotion datasets and five languages, producing a broadly consistent emotional manifold. The authors further develop a learned steering module that can intervene on hidden states to shift the model’s internal emotional perception toward target emotions while largely preserving semantic content; evaluations report strong post-steering classification accuracy for many basic emotions across models and languages. The study combines alignment metrics (cosine similarity, regression error), distortion/stress diagnostics, linear probes, and qualitative rewriting examples to characterize representation geometry, cross-domain robustness, and steerability.

**Strengths:**

- Identification of a low-dimensional, directionally encoded emotional manifold: The paper demonstrates that emotions in LLMs occupy a low-dimensional subspace that is interpretable and directionally organized across layers, with principal axes (PC1–PC3) showing high rank correlations in many models/layers.
- Cross-dataset and multilingual generalization of emotional structure: Using eight emotion datasets spanning five languages and diverse textual styles, the authors show that the extracted emotional subspace generalizes (low alignment distortion, above-chance linear probe accuracy), supporting the existence of a near-universal affective subspace in multiple LLM families.
- A learned intervention/steering module that controls internal emotional representations: They introduce and evaluate a module that shifts hidden states toward target emotions. Post-steering emotion prediction rates typically rise substantially (often >85% for many emotions), while semantic-similarity loss remains low, indicating control without wholesale semantic degradation. The method is evaluated across model families and languages, with ablations in the appendix.

**Weaknesses:**

- Geometry vs. local distortion — inconsistent relational preservation: Although global alignment measures (cosine, regression) are often strong, stress and distortion analyses reveal notable local warping of relative geometry in many layers and datasets. Thus the emotional manifold is not uniformly faithful to human emotion-space relations, which complicates interpretation and downstream use.
- Uneven multilingual and dataset robustness: Performance and steerability degrade in lower-resource settings (e.g., some emotions in Hindi/Bhaav), and certain datasets (e.g., Go-Emotions) show high layer-wise distortion. This suggests lexical sparsity, annotation imbalance, or domain mismatch limit universality claims and practical applicability across all languages/styles.
- Potential semantic and safety/ethical concerns with steering: Although semantic-similarity loss is reported low, steering produces surface rewrites that can alter tone, register, or pragmatics (examples show forceful rewrites for anger). The paper does not deeply address possible misuse (manipulating perceived emotion), downstream impacts on user trust, or safeguards for safe deployment. Additionally, steering effectiveness varies across emotions and models; some target emotions remain difficult to induce reliably.

**Questions:**

SEE WEAKNESS

---

> ### Author Response · Authors · 2025-11-13
> **Rebuttal by Authors**
>
> We would like to thank the reviewer for their time and consideration of our submission. We respond to each weakness point-by-point:
>
>
> Response to Weakness 1 (Geometry vs. Local Distortion):
>
>
> We appreciate the reviewer’s attention to this point. This limitation is explicitly acknowledged in the paper (Lines 279–287):
> “Yet the stress and distortion metrics reveal that, within the same models, relational structure can still be substantially warped… This discrepancy reflects the fact that centroidal and regression-based measures capture global alignment, whereas stress and distortion expose finer-grained deviations… Thus, high apparent alignment at the aggregate level can coexist with local irregularities… however, linear probing shows that these spaces can still usefully and predictably predict emotion, even if there is some distortion between the synthetic and human-written spaces.”
> Our claims are directional, not isometric: emotions occupy a low-dimensional, directionally organized subspace, but we do not assert uniform preservation of all pairwise emotional distances across datasets or layers. The local warping revealed by stress/distortion metrics is entirely consistent with high-dimensional representational compression, and is one reason we report a suite of metrics (alignment, probe accuracy, distortion) rather than relying on any single notion of geometric fidelity.
> Importantly, despite these distortions, the emotional directions remain predictive and manipulable. The fact that the same subspace supports successful causal interventions, steering hidden states toward target emotions while preserving semantics, demonstrates that the derived manifold is functionally coherent even where local geometry diverges from human affective structure.
> We will make this clarification more explicit to avoid any impression that perfect relational preservation is being claimed.
>
>
> Response to Weakness 2 (Uneven Multilingual Robustness):
>
>
> We agree that multilingual robustness varies, and we view this as primarily dataset- and domain-driven, not a limitation of the method.
> The German dataset consists of 19th-century German plays, whose archaic register and stylistic patterns are far from both the synthetic data used to construct the manifold and the contemporary German corpora on which LLMs are typically pretrained. The elevated distortion therefore reflects domain mismatch, not linguistic under-representation. Notably, steering remains effective and probe accuracy consistently exceeds chance.
> Hindi, in contrast, is well known to have low-resource representation in most LLMs, which affects the encoding of higher-order semantic and affective features. Even so, our approach still recovers a coherent emotional subspace with usable directional structure, reasonable distortion/stress scores, and substantial steering gains (≈50% absolute improvement in classification). We view this as evidence of robustness under challenging conditions.
> Our universality claim is thus statistical and directional, not absolute across all resource levels or stylistic domains. We will add a brief limitations paragraph clarifying this.
>
>
> Response to Weakness 3 (Semantic/Safety Considerations):
>
>
> We appreciate the reviewer raising this concern. We will add a concise discussion paragraph addressing appropriate use, potential misuse, and deployment considerations, as well as the observation that certain target emotions (e.g., anger) produce stronger stylistic shifts. The steering module operates only on intermediate hidden states and is intended for controlled, opt-in stylistic modulation; we will make this explicit.
>
> Proposed Additional Paragraph:
>
>
> Because our method enables controlled shifts in a model’s internal affective representation, we acknowledge the need to articulate its appropriate use and its limitations. The steering mechanism is intentionally constrained: it acts only on intermediate hidden states, preserves semantic content through an explicit alignment regularizer, and cannot induce or amplify harmful content beyond what the base model already permits. Nonetheless, certain emotions, especially high-arousal or interpersonal ones such as anger or contempt, may yield more forceful stylistic rewrites, reflecting asymmetries in how models encode these emotions. We therefore recommend applying steering only in transparent, user-directed contexts, such as tone adjustment, therapeutic or reflective writing tools, accessibility interfaces, or cross-emotion normalization for evaluation. The method is not intended for covert style manipulation, persuasion, or emotionally charged rewriting without user consent. Finally, variability in steerability across languages and domains (e.g., low-resource or archaic corpora) functions as an inherent boundary: the technique does not uniformly override model behavior but respects representational limits.

---

> ### Author Response · Authors · 2025-11-18
>
> Thank you again for your constructive feedback. I’ve revised the paper to address your comments:
>
> – The geometry vs. local distortion point you highlighted has been clarified in Lines 315–328.
>
> – The concerns regarding multilingual balance and dataset robustness are now addressed in Lines 474–486 and 512–522.
>
> – A concluding paragraph discussing safety and ethical considerations has been added in Lines 523–535.

---

> ### Author Response · Authors · 2025-11-26
>
> Given your feedback on weaknesses, we have made changes to address them in the paper. With these changes and those from the other reviewers, the paper is stronger and many ICLR attendees will likely benefit from thie presentation of this paper.
>
> Would it be possible to reconsider and increase your score?

---

### Meta-Review · Area_Chair_7SNn · 2026-01-08

**Summary:**

This is a borderline case.

The paper receives 4 high-quality reviews, with 2 negative initial ratings (4, 4) and 2 positive initial rating (6, 6).

Reviewer hPou (rating: 4) has not participated in the discussions. This reviewer has concerns in relationship between global alignment and local geometric distortion, multilingual robustness, safety/ethical concerns with steering. The authors have given detailed responses to these concerns.

Reviewer 3kjH (rating: 4) has participated in the discussions. Most of this reviewer's concerns have been addressed. Reviewer 3kjH and the authors are discussing on left problem, which may be addressed if the discussions continue.

Another two reviewers give positive initial ratings and the authors have given detailed responses and additional results to address the reviewers' concerns.

**Reviewer Concerns:**

See the above comments.

**Reviewer Scores:**

See the above comments.

---

### Decision · Program_Chairs · 2026-01-26

Accept (Poster)